# PHYSICS-ASSISTED AND TOPOLOGY-INFORMED DEEP LEARNING FOR WEATHER PREDICTION

## ABSTRACT

Weather prediction is crucial for decision-making in various social and economic sectors. The classical numerical weather prediction methods cannot incorporate the historical observations to enhance the underlying physical models, whereas the existing data-driven, deep learning-based weather prediction methods disregard either the **physics** of the weather evolution or the **topology** of the Earth's surface. In light of these disadvantages, we develop PASSAT, a novel Physics-ASSisted And Topology-informed deep learning model for weather prediction. PASSAT attributes the weather evolution to two key factors: (i) the advection process that can be characterized by the advection equation and the Navier-Stokes equation; (ii) the Earth-atmosphere interaction that is difficult to both model and calculate. PASSAT also takes the topology of the Earth's surface into consideration, other than simply treating it as a plane. Therefore, PASSAT numerically solves the advection equation and the Navier-Stokes equation on the spherical manifold, utilizes a spherical graph neural network to capture the Earth-atmosphere interaction, and generates the initial velocity fields that are critical to solving the advection equation, from the same spherical graph neural network. These building blocks constitute a deep learning-based, **physics-assisted** and **topology-informed** weather prediction model. In the $5.625°$-resolution ERA5 data set, PASSAT outperforms both the state-of-the-art deep learning-based weather prediction models and the operational numerical weather prediction model IFS T42.

## 1 INTRODUCTION

Weather prediction is of paramount importance to social security and economic development, and has attracted extensive research efforts since the ancient time. Among the modern weather prediction methods, numerical weather prediction (NWP) is built upon differential equations that govern the weather evolution (Randall et al., 2007; Bauer et al., 2015). These differential equations attribute the weather evolution to the **advection process** and the **Earth-atmosphere interaction** (Rood, 1987; Smith et al., 1990), as shown in Figure 1. The advection process is the evolution of weather variables (described by the advection equation) driven by the evolution of their velocity fields (described by the Navier-Stokes equation). The Earth-atmosphere interaction encompasses other complex physical processes in the atmosphere, such as radiation, clouds, and subgrid turbulent motions. One particular challenge in NWP is that the Earth-atmosphere interaction is difficult to model and calculate, forming a bottleneck of improving the accuracy of NWP (Hourdin et al., 2017; Kochkov et al., 2024). Besides, the accuracy of NWP does not improve with the increasing amount of historical observations.

On the other hand, data-driven methods that predict the weather based on the historical observations, especially deep learning models, have become very popular in recent years (Ren et al., 2021). With the aid of high-quality and ever-accumulating data, state-of-the-art deep learning models have demonstrated great potentials and been integrated into the modern weather prediction systems. Besides, deep learning-based models are able to remarkably shorten the time consumption in the prediction stage (Bi et al., 2023; Lam et al., 2023). However, these models disregard either the **physics** of the weather evolution or the **topology** of the Earth's surface. Thus, their predictions are often unreliable due to the lack of the physical constraints or suffer from the distortions caused by the topological structure (Schultz et al., 2021).

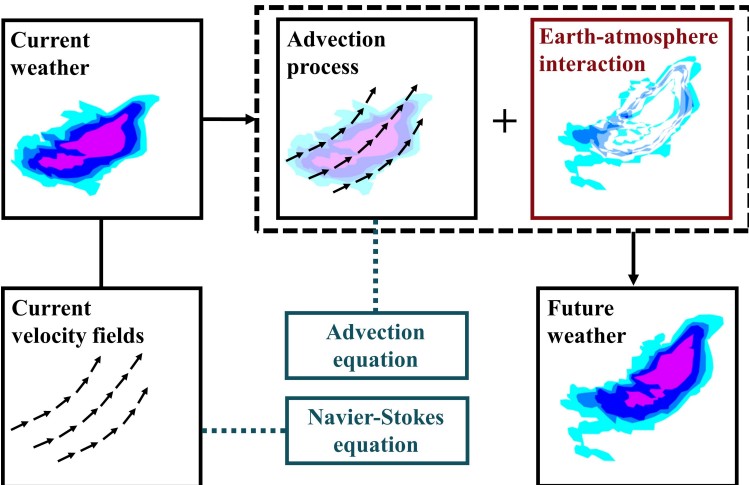

Figure 1: **Attributions of the weather evolution.**

## 1.1 ENHANCING DEEP LEARNING WITH DIFFERENTIAL EQUATIONS

Combining with the differential equations that characterize the weather evolution can enhance the precisions, efficiency and robustness of deep learning models, as the differential equations provide valuable prior knowledge (Xiang et al., 2022). Some works incorporate differential equations into losses during training deep learning models (Daw et al., 2021; de Bezenac et al., 2018). However, tuning weights for the differential equations and computing stochastic gradients of the losses bring new challenges. Some other works utilize deep learning models to correct NWP models (Kwa et al., 2023; Arcomano et al., 2022; Kochkov et al., 2024). Though having high accuracies, these approaches are computationally demanding since they need to both solve all differential equations and train end-to-end neural networks. The closest to ours are (Verma et al., 2024; Zhang et al., 2023), in which neural networks are trained with the aid of differential equations. However, they both overlook the Navier-Stokes equation that drives the elocution of the velocity fields.

Despite that these physics-assisted deep learning models are harder to train and slower in inference compared to the end-to-end deep learning methods, they significantly enhance the robustness of predictions and demonstrate remarkable potentials (Chen et al., 2018).

## 1.2 TAKING TOPOLOGY OF EARTH'S SURFACE INTO CONSIDERATION

The historical observations used during training most deep learning-based weather prediction models are on planar latitude-longitude grids, other than on the spherical surface of the Earth. Nevertheless, neglecting the Earth's topology introduces remarkable distortions, as shown in Figure 2 (Mai et al., 2023; Cohen et al., 2018). For example, the points that are close to the poles turn to be denser on the spherical manifold than on the planar latitude-longitude grid. A notable consequence is that one weather pattern appears differently on the sphere and the plane, such that capturing the weather pattern on the plan suffers from distortions. These distortions also affect the patches and convolution kernels, negatively impacting the deep learning models based on convolutional neural networks or vision transformers (Coors et al., 2018). In addition, the velocity fields defined on the planar are significantly distorted when increasing the latitude towards the poles, which will bring biases to the deep learning models that learn the velocity fields (Verma et al., 2024; Zhang et al., 2023).

## 1.3 CONTRIBUTIONS

In this paper, we propose PASSAT, a novel **P**hysics-**ASS**isted **A**nd **T**opology-informed deep learning model for weather prediction. PASSAT attributes the weather evolution to the analytical advection process and the complex Earth-atmosphere interaction. Within the advection process, the evolution of weather variables is driven by the evolution of their velocity fields, and the two are respectively

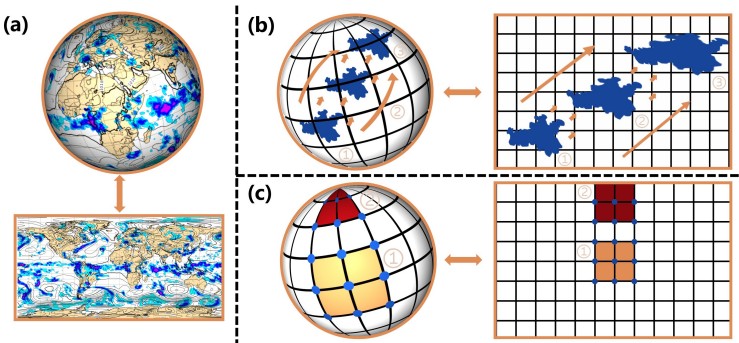

Figure 2: **Distortions due to planar projection.** (a) The spherical and planner representations of the global weather. (b) The same weather patterns on the sphere are distorted on the plane. (c) The convolutions on the sphere are distorted on the plane.

described by the advection equation and the Navier-Stokes equation. PASSAT also takes the topology of the Earth's surface into consideration. Therefore, PASSAT: (i) trains a spherical graph neural network to estimate the Earth-atmosphere interaction; (ii) generates the initial velocity fields with the same spherical graph neural network; (iii) numerically solves the advection equation on the spherical manifold; (iv) updates the velocity fields through numerically solving the Navier-Stokes equation on the spherical manifold. Our contributions are as follows.

- PASSAT seamlessly integrates the historical observations, the physics of the weather evolution and the topology of the Earth's surface, yielding a novel physics-assisted and topology-informed deep learning model for weather prediction.

- Compared to the black-box deep learning models, PASSAT takes advantages of the physical constraints, characterized by the advection equation and the Navier-Stokes equation, and thus remarkably improves the stability of medium-term prediction.

- Compared to the traditional NWP models, PASSAT avoids modeling and calculating the complex Earth-atmosphere interaction. PASSAT is also able to utilize the historical observations to improve the prediction accuracy.

- PASSAT solves the differential equations and trains the graph neural network on the spherical manifold other than on the planar latitude-longitude grid, and thus effectively avoids the distortions brought by the latter.

- We conduct experiments on the 5.625° ERA5 data set, demonstrating the competitive performance of PASSAT compared to the state-of-the-art deep learning models and the NWP model IFS T42.

## 2 RELATED WORKS

**Numerical weather prediction (NWP).** NWP is a fundamental physics-based method for weather prediction (Scher, 2018), utilizing the underlying differential equations to predict how the weather will evolve over the time. For example, the operational Integrated Forecast System (IFS) consists of several NWP models with different spatial resolutions (Bouallègue et al., 2024). Despite of its widespread applications, modeling and calculating the complex Earth-atmosphere interaction are challenging. In addition, solving the differential equations is sensitive to the initial conditions, and also computationally demanding (Kochkov et al., 2024).

**Deep learning-based weather prediction.** Deep learning models learn from the historical observations so as to predict the weather. Although time-consuming during training, deep learning models are rapid during prediction since they do not involve solving the differential equations. State-of-the-art deep learning-based weather prediction models include FourCastNet (Kurth et al., 2023), Pangu (Bi et al., 2023), GraphCast (Lam et al., 2023), ClimaX (Nguyen et al., 2023), Fengwu (Chen et al.,

2023a), and Fuxi (Chen et al., 2023b). Among them, only GraphCast takes the Earth's topology into consideration. However, all of them disregard the underlying physics information.

**Deep learning-based, physics-assisted weather prediction.** Integrating the differential equations with deep learning models significantly improves the precisions, efficiency and robustness of the latter. Notable recent works along this line include ClimODE (Verma et al., 2024) and NowcastNet (Zhang et al., 2023). Different to PASSAT, ClimODE characterizes the evolution of the weather variables with the continuity equation, other than the advection equation. On the other hand, ClimODE updates the velocity fields with a neural network, other than the Navier-Stokes equation. Nowcast-Net focuses on regional precipitation nowcasting, while PASSAT focuses on global, multi-variable and medium-term predictions. Additionally, PASSAT solves the differential equations and trains its graph neural network on a spherical manifold, other than on the planar latitude-longitude grid used by ClimODE and NowcastNet, effectively avoiding the distortions.

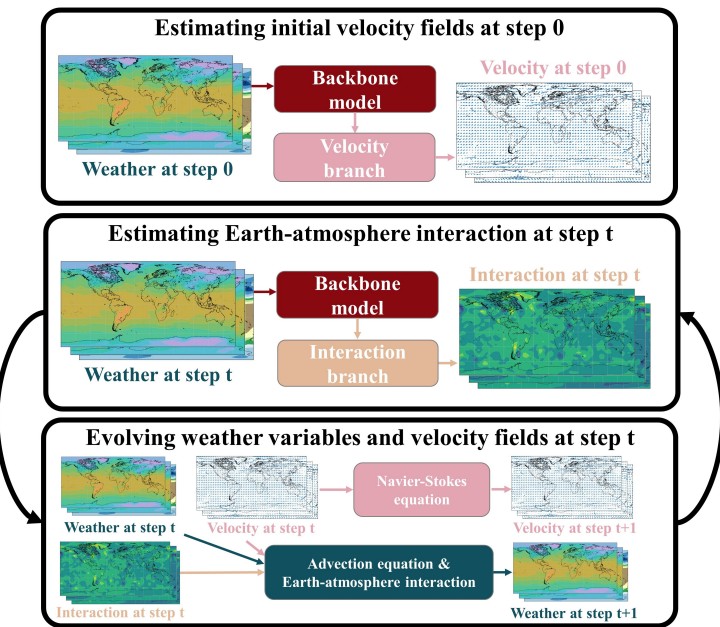

Figure 3: **Overview of PASSAT.**

## 3 METHODS

Considering the attributions of the weather evolution illustrated in Figure 1, we accordingly build a physics-assisted and topology-informed deep learning model for weather prediction, abbreviated as PASSAT. Given any initial time, PASSAT: (i) generates the initial velocity fields of the weather variables with the velocity branch of a spherical graph neural network; and then autoregressively (ii) predicts the effects of the Earth-atmosphere interaction with the interaction branch of the spherical graph neural network; (ii) numerically solves the advection equation on the spherical manifold; (iv) numerically updates the velocity fields through solving the Navier-Stokes equation on the spherical manifold, aided by the initial velocity fields provided by (i). In the following, we will discuss how PASSAT captures the evolution of the weather variables and their velocity fields, via integrating the two differential equations and the spherical graph neural network (see also Figure 3).

We disregard the impact of vertical actions and focus on analyzing the advection equation and the Navier-Stokes equation on the spherical manifold. All the analyses and approaches presented below can be readily extended to scenarios where the vertical actions are taken into account.

We begin by introducing the spherical manifold in Section 3.1 and describing the evolution of the weather variables in Section 3.2. Then, we respectively present the advection equation on the spher-

ical manifold, the spherical graph neural network and the Navier-Stokes equation on the spherical manifold in Section 3.3, Section 3.4 and Section 3.5. Finally, we summarize in Section 3.6.

### 3.1 SPHERICAL MANIFOLD

The historical observations used during training most deep learning-based weather prediction models are on planar latitude-longitude grids, other than on the spherical surface of the Earth. Ignoring this topology information leads to remarkable distortions in both the neural networks and the differential equations. In order to avoid such distortions, we project the weather variables from a planar latitude-longitude grid onto the Earth's surface. Without loss of generality, we assume the Earth's surface to be an ideal unit sphere, with a radius of 1 unit length (6371km).

We denote the unit sphere $\mathbf{S} = \{\mathbf{s} \in \mathbb{R}^3 | \ ||\mathbf{s}||_2 = 1\}$ as the Earth's surface. Any spatial coordinate $\mathbf{s}$ on the unit sphere corresponds to a point $(\phi, \theta)$ within the planar latitude-longitude grid, where $\theta$ is the latitude and $\phi$ is the longitude. Thus, we use $\mathbf{s}$ and $\mathbf{s}(\phi, \theta)$ interchangeably. Given any spatial coordinate $\mathbf{s}$, $\mathbf{e}_\phi(\mathbf{s}) \in \mathbb{R}^3$ and $\mathbf{e}_\theta(\mathbf{s}) \in \mathbb{R}^3$ are two orthogonal unit vectors originated from $\mathbf{s}$ and along the parallel and meridian directions, respectively. We denote $\nabla_\mathbf{s}$ as the spatial gradient on the unit sphere and $\cdot$ as the inner product.

### 3.2 EVOLUTION OF WEATHER VARIABLES

Weather prediction depends on understanding the evolution of weather variables that we are interested in. Given any weather variable $u$, its evolution is characterized as follows.

The weather variable $u$ is viewed as a differentiable, real-valued function $u : T \times \mathbf{S} \to \mathbb{R}$, within which $T$ is the time set and $\mathbf{S}$ is the Earth's surface. According to Figure 1, the evolution of weather variable $u$ is attributed to the advection process and the Earth-atmosphere interaction, as:

$$\frac{\partial u}{\partial t}(t, \mathbf{s}) = (\frac{\partial u}{\partial t}(t, \mathbf{s}))_{\textbf{advection}} + (\frac{\partial u}{\partial t}(t, \mathbf{s}))_{\textbf{interaction}}, \quad \forall (t, \mathbf{s}) \in T \times \mathbf{S}, \tag{1}$$

where $\frac{\partial u}{\partial t}(t, \mathbf{s})$ is the total tendency of weather variable $u$ at time $t$ and spatial coordinate $\mathbf{s}$ and can be decomposed into the tendency due to the advection process, denoted as $(\frac{\partial u}{\partial t}(t, \mathbf{s}))_{\textbf{advection}}$, and the tendency due to the Earth-atmosphere interaction, denoted as $(\frac{\partial u}{\partial t}(t, \mathbf{s}))_{\textbf{interaction}}$.

Once the total tendency $\frac{\partial u}{\partial t}(t, \mathbf{s})$ is known, we can predict the value of $u$ at any future time $t + \Delta t$ according to Newton-Leibniz theorem and using proper numerical methods, as:

$$u(t + \Delta t, \mathbf{s}) = u(t, \mathbf{s}) + \int_t^{t+\Delta t} \frac{\partial u}{\partial t}(\tau, \mathbf{s}) d\tau \approx u(t, \mathbf{s}) + \Delta t \frac{\partial u}{\partial t}(t, \mathbf{s}). \tag{2}$$

Therefore, the key of weather prediction is to compute the tendencies of the advection process and the Earth-atmosphere interaction. Though the tendency of the advection process can be numerically estimated by solving the advection equation on the spherical manifold, the tendency of the Earth-atmosphere interaction is difficult to model and calculate so that we resort to a spherical graph neural network. We introduces them one by one in the following.

### 3.3 ADVECTION EQUATION ON SPHERICAL MANIFOLD

The advection process is the evolution of the weather variables driven by their velocity fields. Given any weather variable $u$, its velocity field $\mathbf{v} : T \times \mathbf{S} \to \mathbb{R}^3$ is a differentiable function of time and spatial coordinate. Since we disregard vertical actions, the velocity field can be express by $\mathbf{v}(t, \mathbf{s}) = v_\theta(t, \mathbf{s})\mathbf{e}_\theta(\mathbf{s}) + v_\phi(t, \mathbf{s})\mathbf{e}_\phi(\mathbf{s})$, where $v_\theta$ and $v_\phi$ are the velocities of $u$ along the meridian and parallel directions, respectively. With particular note, at any initial time $t$ and spacial coordinate $\mathbf{s}$, $u(t, \mathbf{s})$ is known but $\mathbf{v}(t, \mathbf{s})$ is to be calculated.

The tendency of $u$ due to the advection process is given by solving the advection equation (Chandrasekar, 2022), as:

$$(\frac{\partial u}{\partial t}(t, \mathbf{s}))_{\textbf{advection}} + \underbrace{\mathbf{v}(t, \mathbf{s}) \cdot \nabla_\mathbf{s} u(t, \mathbf{s})}_{\text{advective derivative}} = 0, \quad \forall (t, \mathbf{s}) \in T \times \mathbf{S}. \tag{3}$$

Once the advective derivative is known, the tendency of $u$ due to the advection process is known too. On the spherical manifold and the planar latitude-longitude grid, the advective derivative has different forms, and the latter brings distortions in weather prediction, as discussed in the following.

Given a spatial coordinate $\mathbf{s} = \mathbf{s}(\phi, \theta) \in \mathbf{S}$, on the spherical manifold, the advective derivative is in the form of (Lions et al., 1992):

$$\textbf{on spherical manifold:} \quad \mathbf{v}(t, \mathbf{s}) \cdot \nabla_{\mathbf{s}} u(t, \mathbf{s}) = v_\theta(t, \mathbf{s}) \frac{\partial u}{\partial \theta}(t, \mathbf{s}) + \frac{v_\phi(t, \mathbf{s})}{\cos \theta} \frac{\partial u}{\partial \phi}(t, \mathbf{s}). \quad (4)$$

For $v_\theta(t, \mathbf{s})$ and $v_\phi(t, \mathbf{s})$, PASSAT will estimate their initial values utilizing the velocity branch of a spherical graph neural network, and calculate their future values through solving the Navier-Stokes equation. The differentials $\frac{\partial u}{\partial \theta}(t, \mathbf{s})$ and $\frac{\partial u}{\partial \phi}(t, \mathbf{s})$ can be estimated using the difference quotients of $u$ on the planar latitude-longitude grid.

In contrast, on the planar latitude-longitude grid, the advective derivative is in the form of:

$$\textbf{on planar latitude-longitude grid:} \quad \mathbf{v}(t, \mathbf{s}) \cdot \nabla_{\mathbf{s}} u(t, \mathbf{s}) = v'_\theta(t, \mathbf{s}) \frac{\partial u}{\partial \theta}(t, \mathbf{s}) + v'_\phi(t, \mathbf{s}) \frac{\partial u}{\partial \phi}(t, \mathbf{s}), \quad (5)$$

where $v'_\theta(t, \mathbf{s})$ and $v'_\phi(t, \mathbf{s})$ are respectively the velocities along the meridian and parallel directions, but on the latitude-longitude planar grid, not on the spherical manifold. We have $v'_\theta(t, \mathbf{s}) = v_\theta(t, \mathbf{s})$ and $v'_\phi(t, \mathbf{s}) = \frac{v_\phi(t, \mathbf{s})}{\cos \theta}$.

ClimODE and NowcastNet calculate the the advective derivative according to (5), via estimating $v'_\theta(t, \mathbf{s})$ and $v'_\phi(t, \mathbf{s})$ with neural networks. However, we can observe that fixing the value of $v_\phi(t, \mathbf{s})$, $v'_\phi(t, \mathbf{s})$ is not spatial-invariant – it is large when $\mathbf{s}$ is close to the poles and small when $\mathbf{s}$ is close to the equator. Such distortions will affect the pattern recognition of the neural networks. In contrast, PASSAT takes advantages of the spherical manifold, and thus avoids the distortions.

### 3.4 SPHERICAL GRAPH NEURAL NETWORK

As discussed above, to calculate the tendency of $u$ due to the advection process, we need to estimate the initial velocity field of $\mathbf{v}(t, \mathbf{s})$. On the other hand, we need to estimate $(\frac{\partial u}{\partial t}(t, \mathbf{s}))_{\textbf{interaction}}$, the tendency of $u$ due to the Earth-atmosphere interaction. We train a spherical graph neural network to estimate these values.

The spherical graph neural network consists of a backbone model and two branches: the interaction branch that estimates $(\frac{\partial u}{\partial t}(t, \mathbf{s}))_{\textbf{interaction}}$ and the velocity branch that estimates $\mathbf{v}(t, \mathbf{s})$. The spherical graph neural network incorporates the topology information from the spherical manifold, and thus avoids the distortions caused by the planar latitude-longitude grid. For more details, readers are referred to the supplementary material.

Up to now, at time $t$, we have known $(\frac{\partial u}{\partial t}(t, \mathbf{s}))_{\textbf{interaction}}$ (from the spherical graph neural network), as well as $(\frac{\partial u}{\partial t}(t, \mathbf{s}))_{\textbf{advection}}$ (from the advection equation) with the aid of $\mathbf{v}(t, \mathbf{s})$ (from the spherical graph neural network). Therefore, we can the predict the value of $u$ at a future time $t + \Delta t$ according to (2). However, the numerical methods to solve (2) are sensitive to the lead time $\Delta t$. As we will see, the temporal resolution of our weather prediction is 6 hours. When $\Delta t = 6$ hours, the numerical accuracies in solving (2) are acceptable. But when $\Delta t$ becomes larger, these numerical methods are no longer reliable.

To address this issue and enable medium-term or long-term prediction, PASSAT predicts the future $u$ in an autoregressive manner. To be specific, the predicted value of $u$ for $t + 6$ hours will be the initial value of $u$ for predicting $t + 12$ hours; so on and so forth. However, a new challenge arises: the prediction errors originated from the black-box spherical graph neural network accumulate, such that medium-term or long-term prediction will be inevitably biased.

Our remedy is to introduce new physical information to assist PASSAT. To be specific, we no longer trust the velocity field $\mathbf{v}$ estimated by the velocity branch of the spherical graph neural network, except for the initial time $t$. Instead, we solve the Navier-Stokes equation that governs the evolution of the velocity field, to calculate $\mathbf{v}$. We discuss in the following subsection.

### 3.5 NAVIER-STOKES EQUATION ON SPHERICAL MANIFOLD

On the spherical manifold, the velocity field $\mathbf{v}(t, \mathbf{s}) = v_\theta(t, \mathbf{s})\mathbf{e}_\theta(\mathbf{s}) + v_\phi(t, \mathbf{s})\mathbf{e}_\phi(\mathbf{s})$ satisfies the Navier-Stokes equation (Lions et al., 1992):

$$\frac{\partial v_\theta}{\partial t} + \underbrace{(v_\theta \frac{\partial v_\theta}{\partial \theta} + \frac{v_\phi}{\cos\theta} \frac{\partial v_\theta}{\partial \phi})}_{\text{advection}} + \underbrace{v_\phi^2 \tan\theta}_{\text{curvature}} + \underbrace{\frac{1}{\rho} \frac{\partial p}{\partial \theta}}_{\text{pressure gradient force}} + \underbrace{2\omega v_\phi \sin\theta}_{\text{Coriolis force}} + \underbrace{\frac{\mu}{\cos^2\theta} v_\theta}_{\text{viscous friction}} = 0, \quad (6)$$

$$\frac{\partial v_\phi}{\partial t} + \underbrace{(v_\theta \frac{\partial v_\phi}{\partial \theta} + \frac{v_\phi}{\cos\theta} \frac{\partial v_\phi}{\partial \phi})}_{\text{advection}} - \underbrace{v_\phi v_\theta \tan\theta}_{\text{curvature}} + \underbrace{\frac{1}{\rho\cos\theta} \frac{\partial p}{\partial \phi}}_{\text{pressure gradient force}} - \underbrace{2\omega v_\theta \sin\theta}_{\text{Coriolis force}} + \underbrace{\frac{\mu}{\cos^2\theta} v_\phi}_{\text{viscous friction}} = 0, \quad (7)$$

We omit the pair $(t, \mathbf{s})$ for notational simplicity. In the Navier-Stokes equation, $\rho(t, \mathbf{s})$ is the atmospheric density, $\omega = 0.2618$ (radian/hour) is the Earth's rotation speed, $p(t, \mathbf{s})$ is the atmospheric pressure, and $\mu$ is a constant related to the Reynolds constant. For computational efficiency and stability, we simplify the Navier-Stokes equation by retaining only the viscous friction in the Laplacian.

The Navier-Stokes equation governs the evolution of $v_\theta(t, \mathbf{s})$ and $v_\phi(t, \mathbf{s})$. After calculating $\frac{\partial v_\theta(t, \mathbf{s})}{\partial t}$ and $\frac{\partial v_\phi(t, \mathbf{s})}{\partial t}$ from the Navier-Stokes equation, we also apply Newton-Leibniz theorem and numerical methods to predict $v_\theta(t, \mathbf{s})$ and $v_\phi(t, \mathbf{s})$.

### 3.6 SUMMARIZING PASSAT

Here we summarize the prediction procedure of PASSAT. At current time $t$ and spatial coordinate $\mathbf{s}$, for any weather variable $u(t, \mathbf{s})$, we first use the spherical graph neural network to generate its velocity field $(v_\theta(t, \mathbf{s}), v_\phi(t, \mathbf{s}))$. Then, given a temporal resolution $\Delta t$, for $\tau = 0, 1, \cdots$, we iteratively predict the values of $u$ at times $t + (\tau + 1)\Delta t$ in an autoregressive manner. This involves: (i) estimating the tendency of $u$ due to the Earth-atmosphere interaction $(\frac{\partial u}{\partial t}(t + \tau\Delta t, \mathbf{s}))_{\textbf{interaction}}$ using the same spherical graph neural network; (ii) calculating the tendency of $u$ due to the advection process $(\frac{\partial u}{\partial t}(t + \tau\Delta t, \mathbf{s}))_{\textbf{advection}}$ through the advection equation (3) and (4); (iii) calculating the total tendency of $u$ through (1); (iv) calculating the tendency of $v_\theta$ and $v_\phi$ through the Navier-Stokes equation (6) and (7); (v) using Newton-Leibniz theorem and numerical methods to predict the values of $u(t + (\tau + 1)\Delta t, \mathbf{s}), (v_\theta(t + (\tau + 1)\Delta t, \mathbf{s}), v_\phi(t + (\tau + 1)\Delta t, \mathbf{s}))$. In the experiments, we use the Euler's method in the prediction, with $\Delta t = 6$ hours.

## 4 EXPERIMENTS

**Data & Tasks.** The experiments are conducted on the European Centre for Medium-Range Weather Forecasts Reanalysis V5 (ERA5) $5.625°$-resolution data set from 2006 to 2018, provided by WeatherBench (Hersbach et al., 2020; Rasp et al., 2020). The data samples from 2006 to 2015 are used in the training set, 2016 in the validation set, as well as 2017 and 2018 in the test set. The interested weather variables are temperature at 2m height (t2m), temperature at 850hPa pressure level (t), geopotential at 500hPa pressure level (z), u component of wind at 10m height (u10), and v component of wind at 10m height (v10).

We employ PASSAT and the baseline models to predict these weather variables, at a temporal resolution of 6 hours (6am, 12am, 6pm, and 12pm of each day) and lasting for 20 steps (120 hours). The performance metrics are root mean square error (RMSE), anomaly correlation coefficient (ACC) and mean bias error (MBE). Due to the page limit, we only demonstrate the RMSE in this section. For ACC and MBE, as well as more details on data preprocessing, readers are referred to the supplementary material.

We release an open-source Pytorch implementation of PASSAT on PASSAT.

**Baseline deep learning models.** We compare PASSAT with the following baseline deep learning models: (i) ClimODE (Verma et al., 2024); (ii) FourCastNet (Kurth et al., 2023); (iii) Pangu (Bi et al., 2023); (iv) GraphCast (Lam et al., 2023); (v) ClimaX (Nguyen et al., 2023). Among them, ClimaX needs to specify the number of steps (namely, the lead time) in advance, while PASSAT and the other models predict in an autoregressive manner. For fair comparisons, we unify the number of

parameters to the same magnitude (around 1.2 million). We also assign the same weights to different lead times and different weather variables in the losses. We reproduce Pangu and GraphCast as their codes are unavailable in public, while train the other models according to their open-source codes.

We do not compare with NeuralGCM, NowcastNet, Fengwu, and Fuxi, whose codes are unavailable in public too. NeuralGCM requires to solve a complete set of key differential equations, demanding substantial computational resources as the NWP models. NowcastNet exclusively focuses on regional precipitation nowcasting, while we focus on global, multi-variable and medium-term predictions. Fengwu and Fuxi utilize attention-based structures similar to Pangu and Climax, and their focus is on improving long-term predictions via enhancing the training and inference strategies.

**Baseline NWP models.** PASSAT is also compared with the following operational NWP models: (i) IFS T42 (Rasp et al., 2020); (ii) IFS T63 (Rasp et al., 2020). IFS T42 and IFS T63 are the Integrated Forecast System (IFS) model run at two different resolutions, 2.8° and 1.9° respectively. We observe that they are both finer than the 5.625° resolution of PASSAT, at the cost of more computationally demanding in solving the advection equation and the Navier-Stokes equation.

**Training strategy.** We train PASSAT and the baseline deep learning models mentioned above from scratch, following a two-phase approach: pre-training and fine-tuning. In the pre-training phase, we train each model to predict the weather variables for the future 6 hours and the future 12 hours, focusing on the short-term prediction capability. In the fine-tuning phase, we gradually increase the lead time from 18 hours to 72 hours, strengthening the medium-term prediction capability. Such a training strategy is similar to the one adopted in GraphCast (Lam et al., 2023). Note that we will use these models to predict up to 120 hours. For more details on model training, readers are referred to the supplementary material.

Table 1: **Comparisons between PASSAT and the other models in terms of RMSE, over the test set.** We use ∗ to indicate the reproduced GraphCast and Pangu models. For each lead time and each weather variable, the best model RMSE is in **bold** and the second best model RMSE is underlined.

| | Lead Time (h) | PASSAT | GraphCast* | ClimODE | Pangu* | FourCastNet | ClimaX | IFS T42 | IFS T63 |
|---|---|---|---|---|---|---|---|---|---|
| **Physics-assisted** | - | Yes | No | Yes | No | No | No | - | - |
| **Topology-informed** | - | Yes | Yes | No | No | No | No | - | - |
| **Parameters** (M) | - | 1.19 | 1.23 | 1.32 | 1.40 | 1.22 | 1.21 | - | - |
| **t2m** | 24 | 1.18 | **1.16** | 1.72 | 1.73 | 1.72 | 1.74 | - | - |
| | 48 | 1.51 | **1.50** | 2.12 | 1.94 | 1.93 | 2.46 | - | - |
| | 72 | **1.83** | 1.84 | 2.50 | 2.16 | 2.15 | 2.93 | 3.21 | 2.04 |
| | 96 | **2.13** | 2.17 | 2.72 | 2.36 | 2.36 | 4.38 | - | - |
| | 120 | **2.40** | 2.45 | 2.86 | 2.53 | 2.54 | 7.19 | 3.69 | 2.44 |
| **t** | 24 | **1.25** | **1.25** | 2.12 | 1.65 | 1.84 | 1.58 | - | - |
| | 48 | **1.68** | **1.68** | 2.77 | 2.16 | 2.27 | 2.41 | - | - |
| | 72 | 2.14 | 2.16 | 3.35 | 2.64 | 2.69 | 3.14 | 3.09 | **1.85** |
| | 96 | **2.60** | 2.63 | 3.67 | 3.02 | 3.06 | 4.34 | - | - |
| | 120 | 3.00 | 3.04 | 3.84 | 3.31 | 3.34 | 6.39 | 3.83 | **2.52** |
| **z** | 24 | **174.7** | 179.9 | 372.3 | 268.4 | 306.7 | 247.8 | - | - |
| | 48 | **315.9** | 326.0 | 574.8 | 448.4 | 460.5 | 493.8 | - | - |
| | 72 | 446.1 | 458.0 | 739.9 | 589.2 | 587.9 | 684.1 | 489.0 | **268.0** |
| | 96 | **562.5** | 574.2 | 833.1 | 692.0 | 688.3 | 889.5 | - | - |
| | 120 | 660.0 | 670.2 | 890.0 | 762.4 | 761.6 | 1178.0 | 743.0 | **463.0** |
| **u10** | 24 | **1.58** | 1.59 | 2.56 | 1.97 | 2.17 | 2.00 | - | - |
| | 48 | **2.28** | 2.30 | 3.30 | 2.78 | 2.92 | 3.10 | - | - |
| | 72 | **2.92** | 2.94 | 3.87 | 3.38 | 3.48 | 3.72 | - | - |
| | 96 | **3.42** | 3.44 | 4.16 | 3.76 | 3.85 | 4.38 | - | - |
| | 120 | **3.78** | 3.81 | 4.31 | 4.00 | 4.08 | 5.00 | - | - |
| **v10** | 24 | **1.61** | 1.62 | 2.85 | 2.04 | 2.25 | 2.04 | - | - |
| | 48 | **2.32** | 2.34 | 3.51 | 2.85 | 2.98 | 3.21 | - | - |
| | 72 | **2.97** | 3.00 | 4.12 | 3.47 | 3.55 | 3.86 | - | - |
| | 96 | **3.50** | 3.53 | 4.34 | 3.88 | 3.95 | 4.38 | - | - |
| | 120 | **3.90** | 3.93 | 4.43 | 4.13 | 4.21 | 4.54 | - | - |

**Results.** As demonstrated in Table 1, PASSAT outperforms the other deep learning models in most weather variables across different lead times. The closest to PASSAT is GraphCast, which takes the topology of the Earth's surface into consideration. However, GraphCast ignores the physics of the weather evolution, and thus has to use a more complex graph structure than PASSAT (twice in terms of the number of nodes and three times in terms of the number of edges). ClimODE, despite of its physics-assisted structure, does not perform well. This could be attributed to the following reasons: (i) ClimODE characterizes the evolution of the weather variables with the continuity equation, other

than the advection equation; (ii) ClimODE updates the velocity fields with a neural network, other than the Navier-Stokes equation; (iii) ClimODE ignores the topological information, and thus suffers from the distortions. In contrast, PASSAT solves the advection equation on the spherical manifold to estimate the evolution of the weather variables, solves the Navier-Stokes equation on the spherical manifold to update the velocity fields, and trains a spherical graph neural network to estimate the Earth-atmosphere interaction and the initial velocity fields. Therefore, PASSAT benefits from both the physics information and the topology information, allowing it to achieve better performance.

The RMSEs of IFS T42 and IFS T63 are from Rasp et al. (2020), only including t2m, t and z for lead times of 72 and 120 hours. We can observe that PASSAT outperforms IFS T42, a pure physical model solved at a finer resolution ($2.8°$). Improving the resolution of the physical model from $2.8°$ to $1.9°$, IFS T63 surpasses PASSAT and the other deep learning models, nevertheless at the cost of high computational complexity.

Table 2: **Ablation studies in terms of RMSE, over the test set.** For each lead time and each weather variable, the best model RMSE is in **bold** and the second best model RMSE is underlined. We also compare the memories used in fine-tuning the models to predict for the future 72 hours.

| | Lead-Time (h) | PASSAT | w/o Navier-Stokes equation | w/o Navier-Stokes equation and advection equation |
|---|---|---|---|---|
| **Parameters** (M) | - | 1.19 | 1.19 | 1.37 |
| **Memories** (M) | - | 8708 | 10061 | 7387 |
| **t2m** | 24 | 1.18 | **1.17** | 1.21 |
| | 48 | **1.51** | **1.51** | 1.55 |
| | 72 | **1.83** | **1.83** | 1.87 |
| | 96 | **2.13** | 2.14 | 2.16 |
| | 120 | **2.40** | 2.42 | 2.41 |
| **t** | 24 | **1.25** | 1.26 | 1.29 |
| | 48 | 1.68 | **1.67** | 1.73 |
| | 72 | **2.14** | **2.14** | 2.21 |
| | 96 | **2.60** | **2.60** | 2.66 |
| | 120 | **3.00** | **3.00** | 3.05 |
| **z** | 24 | 174.7 | **173.8** | 189.4 |
| | 48 | 315.9 | **313.9** | 336.7 |
| | 72 | 446.1 | **444.2** | 468.1 |
| | 96 | **562.5** | **562.5** | 582.1 |
| | 120 | **660.0** | 661.5 | 675.0 |
| **u10** | 24 | 1.58 | **1.57** | 1.62 |
| | 48 | **2.28** | **2.28** | 2.36 |
| | 72 | **2.92** | **2.92** | 3.00 |
| | 96 | **3.42** | **3.42** | 3.48 |
| | 120 | **3.78** | 3.79 | 3.83 |
| **v10** | 24 | **1.61** | **1.61** | 1.65 |
| | 48 | 2.32 | **2.31** | 2.39 |
| | 72 | 2.97 | **2.96** | 3.05 |
| | 96 | 3.50 | **3.49** | 3.56 |
| | 120 | 3.90 | **3.89** | 3.94 |

## 5 ABLATION STUDIES

As shown in Table 1, PASSAT and GraphCast outperform the other models, demonstrating the superiority of incorporating the topology information. Below, we conduct ablation studies to assess the effectiveness of the physical information used inside PASSAT. First, we assess the effectiveness of the Navier-Stokes equation. To do so, we retrain PASSAT (without the Navier-Stokes equation) by: (i) keeping to numerically solve the advection equation; (ii) keeping to use a spherical graph neural network to learn the earth-atmosphere interaction; (iii) using the velocity branch of the spherical graph neural network, other than the Navier-Stokes equation, to generate the velocity fields for all times. This approach is similar to NowcastNet. Second, we further assess the effectiveness of both the Navier-Stokes equation and the advection equation. To do so, we retrain PASSAT (without the Navier-Stokes equation and the advection equation) by using a spherical graph neural network to

predict the weather in an end-to-end manner, without numerically solving either the Navier-Stokes equation or the advection equation. This approach is similar to GraphCast.

The results are depicted in Table 2. For medium-term prediction, updating the velocity fields by the Naiver-Stokes equation incorporates the physical constraints, and is hence better than updating the velocity fields from the velocity branch of the spherical graph neural network. Besides, it is able to significantly reduce the memory consumption. Further removing the advection equation essentially affects the prediction ability of PASSAT, showing that treating the entire advection process (including the advection equation and the Earth-atmosphere interaction) as a black box is inappropriate. In comparison, PASSAT only treats the Earth-atmosphere interaction as a black box.

## 6 CONCLUSIONS AND FUTURE WORKS

In this paper, we propose PASSAT, a novel physics-assisted and topology-informed deep learning model for weather prediction. PASSAT seamlessly integrates the advection equation and the Navier-Stokes equation that govern the evolution of the weather variables and their velocity fields, with a graph neural network that estimates the complex Earth-atmosphere interaction and the initial velocity fields. PASSAT also takes the topology of the Earth's surface into consideration, during solving the equations and training the graph neural network. In the $5.625°$-resolution ERA5 data set, PASSAT outperforms both the state-of-the-art deep learning-based weather prediction models and the operational numerical weather prediction model IFS T42.

As future works, we will extend PASSAT in the following aspects: (i) We will enhance PASSAT by incorporating more weather variables. (ii) We will refine PASSAT via training over a data set with a finer resolution. (iii) We will incorporate new forward integration method that is more efficient than the Euler's method, during training and prediction. We expect that PASSAT is able to motivate more research efforts in combining physics, topology and historical observations for weather prediction.

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

Supplementary Material for
Physics-Assisted and Topology-Informed Deep Learning for Weather Prediction

## A    DATA

### A.1    DATA SET

We follow ClimODE to use the European Centre for Medium-Range Weather Forecasts Reanalysis V5 (ERA5) $5.625°$-resolution data set from 2006 to 2018, provided by WeatherBench (Hersbach et al., 2020; Rasp et al., 2020). Table 3 summarizes the weather variables in the data set. For more details, readers are referred to ERA5.

Table 3: **Weather variables in the data set.**

| Long name | Short name | Description | Unit | Levels |
|---|---|---|---|---|
| geopotential | z | Proportional to the height of a pressure level | $m^2s^{-2}$ | 500hPa |
| temperature | t | Temperature | K | 850hPa |
| 2m_temperature | t2m | Temperature at 2m height above surface | K | - |
| 10m_u_component_of_wind | u10 | Wind in longitude-direction at 10m height | $ms^{-1}$ | - |
| 10m_v_component_of_wind | v10 | Wind in latitude-direction at 10m height | $ms^{-1}$ | - |

### A.2    DATA PREPROCESSING

**Normalization.** Since the weather variables have diverse magnitudes, we use their means and standard deviations in 2006 to normalize the entire data set.

**Mapping the latitude-longitude grid to the sphere.** In PASSAT and GraphCast, we need to project a latitude-longitude point $(\theta, \phi)$ onto the unit sphere, as:

$$\mathbf{s}(\theta, \phi) = \left[ \begin{array}{c} \cos\theta\cos\phi \\ \cos\theta\sin\phi \\ \sin\theta \end{array} \right] \in \mathbf{S}. \tag{8}$$

**Constructing time-space embedding.** As in ClimODE, PASSAT uses the time-space embedding as one input, which encompasses the hour in the day, the day in the year and the spatial information. We follow ClimODE to generate the time-space embedding (Verma et al., 2024), as:

$$\mathbf{embed}(t, \mathbf{s}) = (\mathbf{embed}_{\text{time}}(t), \mathbf{embed}_{\text{space}}(\mathbf{s}), \mathbf{embed}_{\text{time}\times\text{space}}(t, \mathbf{s}), \mathbf{lsm}(\mathbf{s}), \mathbf{oro}(\mathbf{s})). \tag{9}$$

Here $t \in [0, 366 \times 24 - 1] \subset \mathbb{N}$ represents the hour that the data sample is in the year,

$$\mathbf{embed}_{\text{time}}(t) = (\sin(2\pi h/24), \cos(2\pi h/24), \sin(2\pi d/366), \cos(2\pi d/366)) \tag{10}$$

with $h = (t \bmod 24)$ being the hour that the data sample is in the day and $d = [t/24]$ being the day that the data sample is in the year,

$$\mathbf{embed}_{\text{space}}(\mathbf{s}) = (\sin(\theta), \cos(\theta), \sin(\phi), \cos(\phi), \sin(\theta)\cos(\phi), \sin(\theta)\sin(\phi)) \tag{11}$$

with $\mathbf{s} = \mathbf{s}(\theta, \phi)$, and

$$\begin{aligned} \mathbf{embed}_{\text{time}\times\text{space}}(t, \mathbf{s}) &= \mathbf{embed}_{\text{time}}(t) \otimes \mathbf{embed}_{\text{space}}(\mathbf{s}) \\ &= (\sin(2\pi h/24)\sin(\theta), \sin(2\pi h/24)\cos(\theta), ..., \cos(2\pi d/366)\sin(\theta)\sin(\phi)) \end{aligned} \tag{12}$$

with $\otimes$ being the Kronecker product. Besides, $\mathbf{lsm}(\mathbf{s})$ and $\mathbf{oro}(\mathbf{s})$ are respectively the land-sea binary mask and the height of Earth's surface, given in the $5.625°$-resolution ERA5 data set.

For simplicity, in the following, we use $\mathbf{embed}(t)$ to include the time-space embeddings at time $t$ of all spatial coordinates.

# B PASSAT

## B.1 GENERATING WEATHER PREDICTIONS

We will introduce how PASSAT generates weather predictions with: (i) the advection equation; (ii) the Navier-Stokes equation; (iii) the earth-atmosphere interaction characterized by a spherical graph neural network; (iv) the initial velocity fields generated by the same spherical graph neural network.

At any initial prediction time, denoted as step $0$, PASSAT takes the weather variables at step $-1$, as well as the weather variables and the time-space embedding at step $0$ as the inputs. First, PASSAT uses the spherical graph neural network to generate the initial velocity fields at and only at step $0$. Then, PASSAT estimates the effects of the Earth-atmosphere interaction and numerically solves the advection equation at step $0$, with the aid of the generated initial velocity fields. The summations of the effects of the Earth-atmosphere interaction and the solutions of the advection equation are the tendencies of the weather variables at step $0$. After that, PASSAT obtains the tendencies of the velocity fields at step $0$, through numerically solving the Navier-Stokes equation. With numerical integration, PASSAT updates the weather variables and the velocity fields for step $1$. As such, PASSAT conducts weather predictions for more steps in an autoregressive manner.

---

**Algorithm 1:** PASSAT: Predicting each weather variable $u$ for future $T$ steps

**Input:** $\{u^t | t = -1, 0\}, \{\textbf{embed}(t), t = 0, 1, 2, \cdots, T\}, \lambda$
**Output:** $\{u^t | t = 1, 2, \cdots, T\}, \{\mathbf{v}^t = v_\theta^t \mathbf{e}_\theta + v_\phi^t \mathbf{e}_\phi | t = 0, 1, 2, \cdots, T\}$
**for** $t = 0, 1, 2, \cdots, T - 1$ **do**
  **if** $t = 0$ **then**
    | Estimate initial velocity field: $\mathbf{v}^0 = f_\mathbf{v}(u^{-1}, u^0, \textbf{embed}(0))$
  **end**
  Estimate effect of Earth-atmosphere interaction: $\textbf{phy}^t = f_\textbf{phy}(u^{t-1}, u^t, \textbf{embed}(t))$
  ————Compute tendencies of $u, v_\theta, v_\phi$ at step $t$————
$$
\begin{cases}
\frac{\partial u^t}{\partial t} = -v_\theta^t \frac{\partial u^t}{\partial \theta} - \frac{v_\phi^t}{\cos\theta} \frac{\partial u^t}{\partial \phi} + \textbf{phy}^t \\
\frac{\partial v_\theta^t}{\partial t} = -v_\theta^t \frac{\partial v_\theta^t}{\partial \theta} - \frac{v_\phi^t}{\cos\theta} \frac{\partial v_\theta^t}{\partial \phi} - (v_\phi^t)^2 \tan\theta - \frac{\partial z^t}{\partial \theta} - 2\omega v_\phi^t \sin\theta - \mu \frac{v_\theta^t}{\cos^2\theta} \\
\frac{\partial v_\phi^t}{\partial t} = -v_\theta^t \frac{\partial v_\phi^t}{\partial \theta} - \frac{v_\phi^t}{\cos\theta} \frac{\partial v_\phi^t}{\partial \phi} + v_\phi^t v_\theta^t \tan\theta - \frac{1}{\cos\theta} \frac{\partial z^t}{\partial \phi} + 2\omega v_\theta^t \sin\theta - \mu \frac{v_\phi^t}{\cos^2\theta}
\end{cases}
$$
  ————Update $u^t, v_\theta^t, v_\phi^t$————
$$
\begin{cases}
u^{t+1} = u^t + \Delta t \frac{\partial u^t}{\partial t} \\
\hat{v}_\theta^{t+1} = v_\theta^t + \Delta t \frac{\partial v_\theta^t}{\partial t}; \quad v_\theta^{t+1} = (1 - \lambda)v_\theta^t + \lambda \hat{v}_\theta^{t+1} \\
\hat{v}_\phi^{t+1} = v_\phi^t + \Delta t \frac{\partial v_\phi^t}{\partial t}; \quad v_\phi^{t+1} = (1 - \lambda)v_\phi^t + \lambda \hat{v}_\phi^{t+1}
\end{cases}
$$
**end**

---

We summarize the details in Algorithm 1. For simplicity, we omit the spatial coordinate $\mathbf{s}$, and use $u^t$ to denote $u(t, \mathbf{s})$ and $\mathbf{v}^t$ to denote $\mathbf{v}(t, \mathbf{s})$. We use $f_\mathbf{v}$ and $f_\textbf{phy}$ to denote the velocity and physics branches of the spherical graph neural network, respectively. We use $\textbf{phy}^t$ to denote the estimated effect of the Earth-atmosphere iteration for time $t$ and spatial coordinate $\mathbf{s}$. Within the Navier-Stokes equation, $\frac{1}{\rho}\frac{\partial p^t}{\partial \phi}$ and $\frac{1}{\rho}\frac{\partial p^t}{\partial \theta}$ are unknown. We replace them with the gradients of the geopotential $z^t$ at the 500 hPa pressure level as a substitute, normalized by a factor of 0.01. To ensure the stability of the estimated velocity fields, we output a convex combination of the current velocity field $(v_\theta^t, v_\phi^t)$ and the intermediately estimated velocity field $(\hat{v}_\theta^t, \hat{v}_\phi^t)$, with a parameter $\lambda$ that is set as $\frac{1}{60}$ in the experiments.

## B.2 STRUCTURE

In Table 4, we illustrate the structures of PASSAT and its variant. For PASSAT without the advection and Navier-Stokes equations, we no longer need the velocity branch, and merge the physics branch with the backbone. For PASSAT without the Navier-Stokes equation, its structure remains the same as PASSAT and we do not list here.

Table 4: **The structure of PASSAT.** The number in the brackets is the output dimension.

| | PASSAT | w/o advection and Navier-Stokes equations |
|---|---|---|
| **Num Nodes** | 2048 | 2048 |
| **Num Edges** | 20480 | 20480 |
| **Input Embedding** | 1× MLP (48) | 1× MLP (48) |
| **Backbone** | 2× Basic Block (48) | [2× Basic Block (48), 1× Basic Block (96)] |
| **Velocity Branch** | 1× Basic Block (48) | - |
| **Physics Branch** | [2× Basic Block (48), 1× Basic Block (24)] | - |
| **Parameters** | 1.19 (M) | 1.37 (M) |

### B.2.1 GRAPH IN PASSAT

The spherical graph neural network of PASSAT is denoted as $(\mathcal{N}, \mathcal{E}, A)$, where $\mathcal{N}$ is the node set, $\mathcal{E}$ is the edge set and $A$ is the adjacency matrix.

Each node of PASSAT is represented by a spatial coordinate. Corresponding to the $5.625°$-resolution data set, there are 2048 nodes, represented as:

$$(\mathbf{n}_0, \mathbf{n}_1, \cdots, \mathbf{n}_{63}, \cdots, \mathbf{n}_{2047}) = (\mathbf{s}(\phi_0, \theta_0), \mathbf{s}(\phi_1, \theta_0), \cdots, \mathbf{s}(\phi_{63}, \theta_0), \cdots, \mathbf{s}(\phi_{63}, \theta_{31})). \quad (13)$$

The $(i, j)$-th element of the original adjacency matrix $A$ is given by:

$$(A)_{ij} = \exp(-||\mathbf{n}_i - \mathbf{n}_j||_2^2). \quad (14)$$

To improve the computational efficiency, we prune $A$ by setting its elements to zero if they are below a given threshold. We set two thresholds, so that in the pruned adjacency matrices, the sparsest row has 6 and 10 non-zeros, respectively. We denote the pruned adjacency matrices as $A_5$ and $A_9$, respectively. We also re-normalize them to avoid exploding/vanishing gradients (Kipf & Welling, 2016), as:

$$A_k \Rightarrow D_k^{-\frac{1}{2}} A_k D_k^{-\frac{1}{2}}, \quad k \in \{5, 9\}, \quad (15)$$

where $D_k$ is a diagonal degree matrix, with $(D_k)_{ii} = \sum_j (A_k)_{ij}$.

We will use both $A_5$ and $A_9$ for feature extraction. In addition, we define the edge set $\mathcal{E}$ by $A_5$. If the $(i, j)$-th element of $A_5$ is non-zero, we say that nodes $\mathbf{n}_i$ and $\mathbf{n}_j$ are neighbors, and $(\mathbf{n}_i, \mathbf{n}_j) \in \mathcal{E}$.

### B.2.2 INITIAL STATES OF NODES AND EDGES

As shown in Algorithm 1, at step 0, the input of PASSAT includes $\mathbf{u}^{-1}$, $\mathbf{u}^0$ and **embed**(0), in which we use $\mathbf{u}$ to stack all weather variables. These three terms determine the initial states of nodes and edges. For node $\mathbf{n}_i$, that is:

$$\textbf{Initialization:} \quad h_i^{\text{node}} = (\mathbf{u}^{-1}(\mathbf{n}_i), \mathbf{u}^0(\mathbf{n}_i), \textbf{embed}(0, \mathbf{n}_i)), \quad (16)$$

where $\mathbf{u}^0(\mathbf{n}_i)$ and $\mathbf{u}^0(\mathbf{n}_i)$ denote the weather variables of spatial coordinate $\mathbf{n}_i$ at step $-1$ and step 0, respectively. For edge $(\mathbf{n}_i, \mathbf{n}_j)$ connecting nodes $\mathbf{n}_i$ and $\mathbf{n}_j$, its initial state is

$$\textbf{Initialization:} \quad h_{ij}^{\text{edge}} = (\mathbf{n}_i - \mathbf{n}_j, ||\mathbf{n}_i - \mathbf{n}_j||_2). \quad (17)$$

### B.2.3 MAIN BLOCKS IN PASSAT

There exist two main blocks within PASSAT: the input embedding block and the basic block. The basic block further consists of graph connection and graph convolution, as shown in Figure 4.

In graph connection, the edge connection involves concatenating the hidden states of the two nodes $\mathbf{n}_i$ and $\mathbf{n}_j$ connected by each edge $(\mathbf{n}_i, \mathbf{n}_j)$, written as:

$$h_{ij}^{\text{edge}} \Rightarrow (h_{ij}^{\text{edge}}, h_i^{\text{node}}, h_j^{\text{node}}), \quad (18)$$

where $h_{ij}^{\text{edge}}$ is the hidden state of edge $(\mathbf{n}_i, \mathbf{n}_j)$, while $h_i^{\text{node}}$ and $h_j^{\text{node}}$ are the hidden states of nodes $\mathbf{n}_i$ and $\mathbf{n}_j$, respectively. The node connection is to concatenate the sum of the hidden states of edges that each node $\mathbf{n}_i$ connects, written as:

$$h_i^{\text{node}} \Rightarrow (h_i^{\text{node}}, \sum_{j:(\mathbf{n}_i, \mathbf{n}_j) \in \mathcal{E}} h_{ij}^{\text{edge}}). \tag{19}$$

The graph convolution uses the adjacency matrices $A_5$ and $A_9$ to aggregate the hidden states of all nodes, and concatenate the results to update the hidden states of all nodes, written as:

$$h_i^{\text{node}} \Rightarrow (h_i^{\text{node}}, \sum_j (A_5)_{ij} h_j^{\text{node}}, \sum_j (A_9)_{ij} h_j^{\text{node}}). \tag{20}$$

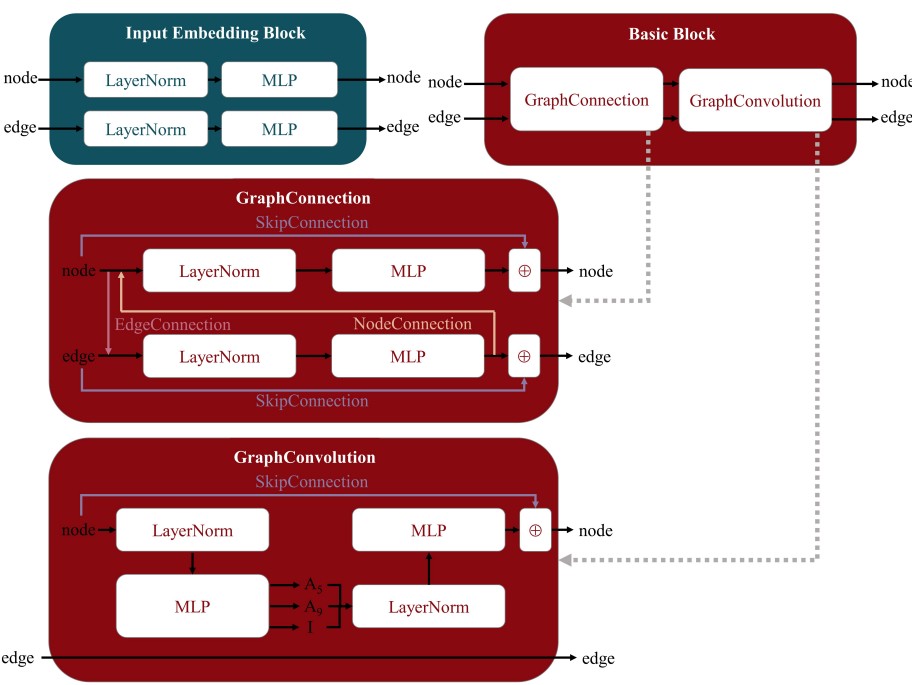

Figure 4: **Main blocks in PASSAT.**

## C  PERFORMANCE METRICS

**Root mean square error (RMSE).** Given a weather variable $u$, we suppose that the initial time is $0$ and that the lead time is $\tau$. The RMSE is defined as:

$$\text{RMSE}(\tau) = \sqrt{\frac{1}{|\mathbf{S}_d|} \sum_{\mathbf{s} \in \mathbf{S}_d} a_{\mathbf{s}} (u(\tau, \mathbf{s}) - \check{u}(\tau, \mathbf{s}))^2}. \tag{21}$$

Therein, $\mathbf{S}_d \subset \mathbf{S}$ is the set of discrete spatial coordinates, while $u(\tau, \mathbf{s})$ and $\check{u}(\tau, \mathbf{s})$ are the prediction and observation of $u$ at lead time $\tau$ and spatial coordinate $\mathbf{s}$. The weight $a(\mathbf{s})$ is defined as:

$$a(\mathbf{s}) = \frac{\cos\theta}{\frac{1}{|\mathbf{S}_d|} \sum_{\mathbf{s}' \in \mathbf{S}_d} \cos\theta'}. \tag{22}$$

The reported RMSE is the average over all initial times.

**Anomaly correlation coefficient (ACC).** Given a weather variable $u$, we suppose that the initial time is 0 and that the lead time is $\tau$. The ACC is defined as:

$$\text{ACC}(\tau) = \frac{\sum_{\mathbf{s} \in \mathbf{S}_d} a_{\mathbf{s}} \text{Clim}(u(\tau, \mathbf{s})) \text{Clim}(\check{u}(\tau, \mathbf{s}))}{\sqrt{\sum_{\mathbf{s} \in \mathbf{S}_d} a_{\mathbf{s}} \text{Clim}(u(\tau, \mathbf{s}))^2 \times \sum_{\mathbf{s} \in \mathbf{S}_d} a_{\mathbf{s}} \text{Clim}(\check{u}(\tau, \mathbf{s}))^2}}. \tag{23}$$

Therein, we define:

$$\text{Clim}(u(\tau, \mathbf{s})) = u(\tau, \mathbf{s}) - C(\mathbf{s}) - \frac{1}{|\mathbf{S}_d|} \sum_{\mathbf{s}' \in \mathbf{S}_d} (u(\tau, \mathbf{s}') - C(\mathbf{s}')), \tag{24}$$

where $C(\mathbf{s})$ is the climatological mean of weather variable $u$ at spatial coordinate $\mathbf{s}$, computed using the ERA5 data set of 2006. The reported ACC is the average over all initial times.

# D TRAINING DETAILS

## D.1 LOSS FUNCTIONS

Given a weather variable $u$ and at any initial time denoted by 0, the loss function of ClimODE, GraphCast, Pangu, FourCastNet, and ClimaX is given by:

$$\mathcal{L}_{\text{basic}} = \frac{1}{T_a} \sum_{\tau \in 1:T_a} \frac{1}{|\mathbf{S}_d|} \sum_{\mathbf{s} \in \mathbf{S}_d} (u(\tau, \mathbf{s}) - \check{u}(\tau, \mathbf{s}))^2. \tag{25}$$

Therein, $\mathbf{S}_d \subset \mathbf{S}$ is the set of discrete spatial coordinates, while $u(\tau, \mathbf{s})$ and $\check{u}(\tau, \mathbf{s})$ are the prediction and observation of $u$ at lead time $\tau$ and spatial coordinate $\mathbf{s}$. We use $T_a$ to denote the number of autoregressive steps. Then, $\mathcal{L}_{\text{basic}}$ is averaged over all weather variables and all initial times.

The predictions of PASSAT also involve the initial velocity fields, whose values must be controlled. Therefore, we introduce several penalty terms to the loss function. Given a velocity field $v$ and at any initial time denoted by 0, we define:

$$\mathcal{L}_{\text{velocity}} = \mathcal{L}_{\text{velocity}}^1 + \mathcal{L}_{\text{velocity}}^2 + \mathcal{L}_{\text{velocity}}^3, \tag{26}$$

where

$$\mathcal{L}_{\text{velocity}}^1 = \frac{\lambda_1}{2|\mathbf{S}_d|} \sum_{\mathbf{s} \in \mathbf{S}_d} [(v_\theta(0, \mathbf{s}))^2 + (v_\phi(0, \mathbf{s}))^2], \tag{27}$$

$$\mathcal{L}_{\text{velocity}}^2 = \frac{\lambda_2}{2|\mathbf{S}_d|} \sum_{\mathbf{s} \in \mathbf{S}_d} [(\frac{\partial}{\partial \theta} v_\theta(0, \mathbf{s}))^2 + (\frac{\partial}{\partial \theta} v_\phi(0, \mathbf{s}))^2], \tag{28}$$

$$\mathcal{L}_{\text{velocity}}^3 = \frac{\lambda_3}{2|\mathbf{S}_d|} \sum_{\mathbf{s} \in \mathbf{S}_d} [(\frac{\partial}{\partial \phi} v_\theta(0, \mathbf{s}))^2 + (\frac{\partial}{\partial \phi} v_\phi(0, \mathbf{s}))^2]. \tag{29}$$

Therein, $\lambda_1 = 10$, $\lambda_2 = 0.1$ and $\lambda_3 = 0.1$ are constants to penalize the initial velocity field and its smoothness. Then, $\mathcal{L}_{\text{velocity}}$ is averaged over all velocity fields and all initial times.

In summary, the loss function of PASSAT is given by $\mathcal{L} = \mathcal{L}_{\text{basic}} + \mathcal{L}_{\text{velocity}}$.

## D.2 TRAINING STRATEGIES

We train PASSAT and the baseline deep learning models from scratch in an autoregressive manner (except for ClimaX); that is, we treat the current predictions as observations and feed them back to the models to generate future predictions.

The training of all models follows a two-phase approach: pre-training and fine-tuning (Lam et al., 2023). In the pre-training phase, we train each model to predict the weather variables for the future 6 hours and the future 12 hours, focusing on the short-term prediction capability. In the fine-tuning phase, we gradually increase the lead time from 18 hours to 72 hours, for the sake of strengthening the medium-term prediction capability.

We use the AdamW optimizer with parameters $\beta_1 = 0.9$ and $\beta_2 = 0.999$, and set the weight decay as 0.05. Gradient clipping is also employed, with a maximum norm value of 5. We use PyTorch for training, validation and prediction, on four GeForece RTX 2080.

In the pre-training phase, the batch size is 8 per GPU (32 in total). In the fine-tuning phase, the batch size is 2 per GPU (8 in total) to prevent memory overflow. In the pre-training phase, we adjust the learning rate with the Cosine-LR-Scheduler, and set the maximum and minimum learning rates to 1e-3 and 3e-7, respectively. In the fine-tuning phase, the learning rate is 7.5e-8.

# E  STRUCTURES OF BASELINE DEEP LEARNING MODELS

We consider the following baseline deep learning models: GraphCast, Pangu, ClimODE, FourCast-Net, and ClimaX. For fair comparisons, we unify the number of parameters to the same magnitude (around 1.2 million). In Tables 5–9, we show their modified structures.

Table 5: **The modification of GraphCast.**

|  | Original | Modified (this paper) |
|---|---|---|
| **Num Nodes** | 1079202 | 4096 |
| **Num Edges** | 5061126 | 61440 |
| **Num Embedding Layers** | 5 | 1 |
| **Embedding Dimension** | 512 | 48 |
| **Num Multi-mesh Layers** | 16 | 5 |
| **Parameters** | 36.7 (M) | 1.23 (M) |

Table 6: **The modification of Pangu.**

|  | Original | Modified (this paper) |
|---|---|---|
| **Embedding Dimension** | 192 | 72 |
| **Num Layers** | [2(192), 6(384), 6(384), 2(192)] | [2(72), 2(144), 2(144), 2(72)] |
| **Num Heads** | [6, 12, 12, 6] | [4, 4, 4, 4] |
| **Patch Sizes** | (2, 4, 4) | (1, 1) |
| **Window Sizes** | (2, 6, 12) | (4, 4) |
| **Parameters** | 256 (M) | 1.40 (M) |

Table 7: **The modification of ClimODE.**

|  | Original | Modified (this paper) |
|---|---|---|
| **Num Layers (Velocity)** | [5(128), 3(64), 2(10)] | [5(64), 3(32), 2(10))] |
| **Num Layers (Noise)** | [3(128), 2(64), 2(10)] | [4(96), 2(64), 2(5)] |
| **Parameters** | 2.8 (M) | 1.32 (M) |

# F  COMPARISONS WITH BASELINE MODELS

We compare the ACCs of PASSAT and the other models in Table 10. Similar to the comparisons of RMSEs, PASSAT is the best among all deep learning models and Graphcast is the closest one. IFS T63 outperforms IFS T42 and all deep learning models, but is computationally expensive.

Table 8: **The modification of FourCastNet.**

|  | Original | Modified (this paper) |
|---|---|---|
| **Embedding Dimension** | 768 | 128 |
| **Num Layers** | 12 | 8 |
| **Num Blocks** | 16 | 8 |
| **Parameters** | 59.1 (M) | 1.22 (M) |

Table 9: **The modification of ClimaX.**

|  | Original | Modified (this paper) |
|---|---|---|
| **Embedding Dimension** | 1024 | 128 |
| **Num Layers** | 8 | 6 |
| **Num Decoder Layers** | 2 | 2 |
| **Num Heads** | 16 | 8 |
| **Parameters** | 107 (M) | 1.36 (M) |

Table 10: **Comparisons between PASSAT and the other models in terms of ACC, over the test set.** We use ∗ to indicate the reproduced GraphCast and Pangu models. For each lead time and each weather variable, the best model ACC is in **bold** and the second best model ACC is underlined.

|  | Lead-Time (h) | PASSAT | GraphCast* | ClimODE | Pangu* | FourCastNet | ClimaX | IFS T42 | IFS T63 |
|---|---|---|---|---|---|---|---|---|---|
| **Manifold-informed** | - | Yes | Yes | No | No | No | No | - | - |
| **Physical-informed** | - | Yes | No | Yes | No | No | No | - | - |
| **Parameters** (M) | - | 1.19 | 1.23 | 1.32 | 1.40 | 1.22 | 1.21 | - | - |
| **t2m** | 24 | 0.97 | **0.98** | 0.94 | 0.94 | 0.94 | 0.94 | - | - |
|  | 48 | **0.96** | **0.96** | 0.91 | 0.93 | 0.93 | 0.88 | - | - |
|  | 72 | **0.94** | **0.94** | 0.88 | 0.91 | 0.91 | 0.82 | 0.87 | **0.94** |
|  | 96 | **0.92** | **0.92** | 0.86 | 0.89 | 0.89 | 0.67 | - | - |
|  | 120 | 0.90 | 0.89 | 0.84 | 0.88 | 0.88 | 0.41 | 0.83 | **0.92** |
| **t** | 24 | **0.97** | **0.97** | 0.90 | 0.94 | 0.93 | 0.95 | - | - |
|  | 48 | **0.94** | **0.94** | 0.83 | 0.90 | 0.89 | 0.87 | - | - |
|  | 72 | 0.90 | 0.90 | 0.74 | 0.85 | 0.84 | 0.77 | 0.86 | **0.94** |
|  | 96 | **0.85** | **0.85** | 0.69 | 0.79 | 0.79 | 0.55 | - | - |
|  | 120 | 0.80 | 0.80 | 0.66 | 0.75 | 0.74 | 0.26 | 0.78 | **0.90** |
| **z** | 24 | **0.99** | 0.98 | 0.93 | 0.97 | 0.95 | 0.97 | - | - |
|  | 48 | **0.95** | **0.95** | 0.82 | 0.90 | 0.89 | 0.87 | - | - |
|  | 72 | 0.90 | 0.89 | 0.70 | 0.82 | 0.82 | 0.74 | 0.90 | **0.97** |
|  | 96 | **0.84** | 0.83 | 0.61 | 0.73 | 0.74 | 0.53 | - | - |
|  | 120 | 0.77 | 0.76 | 0.56 | 0.67 | 0.67 | 0.24 | 0.78 | **0.91** |
| **u10** | 24 | **0.93** | **0.93** | 0.80 | 0.88 | 0.85 | 0.88 | - | - |
|  | 48 | **0.84** | 0.83 | 0.61 | 0.75 | 0.72 | 0.67 | - | - |
|  | 72 | **0.72** | 0.71 | 0.43 | 0.60 | 0.57 | 0.45 | - | - |
|  | 96 | **0.60** | 0.59 | 0.34 | 0.48 | 0.46 | 0.14 | - | - |
|  | 120 | **0.49** | 0.48 | 0.28 | 0.39 | 0.38 | 0.03 | - | - |
| **v10** | 24 | **0.92** | **0.92** | 0.75 | 0.87 | 0.84 | 0.87 | - | - |
|  | 48 | **0.83** | **0.83** | 0.54 | 0.73 | 0.70 | 0.64 | - | - |
|  | 72 | **0.71** | 0.70 | 0.31 | 0.57 | 0.55 | 0.39 | - | - |
|  | 96 | **0.58** | 0.57 | 0.22 | 0.43 | 0.43 | 0.09 | - | - |
|  | 120 | **0.46** | 0.45 | 0.19 | 0.33 | 0.33 | 0.04 | - | - |

# G  VISUALIZATIONS

## G.1  MEAN BIAS ERRORS (MBEs)

We visualize the mean bias errors (MBEs) of PASSAT for two lead times: 12 hours and 48 hours, respectively in Figures 5 and 6.

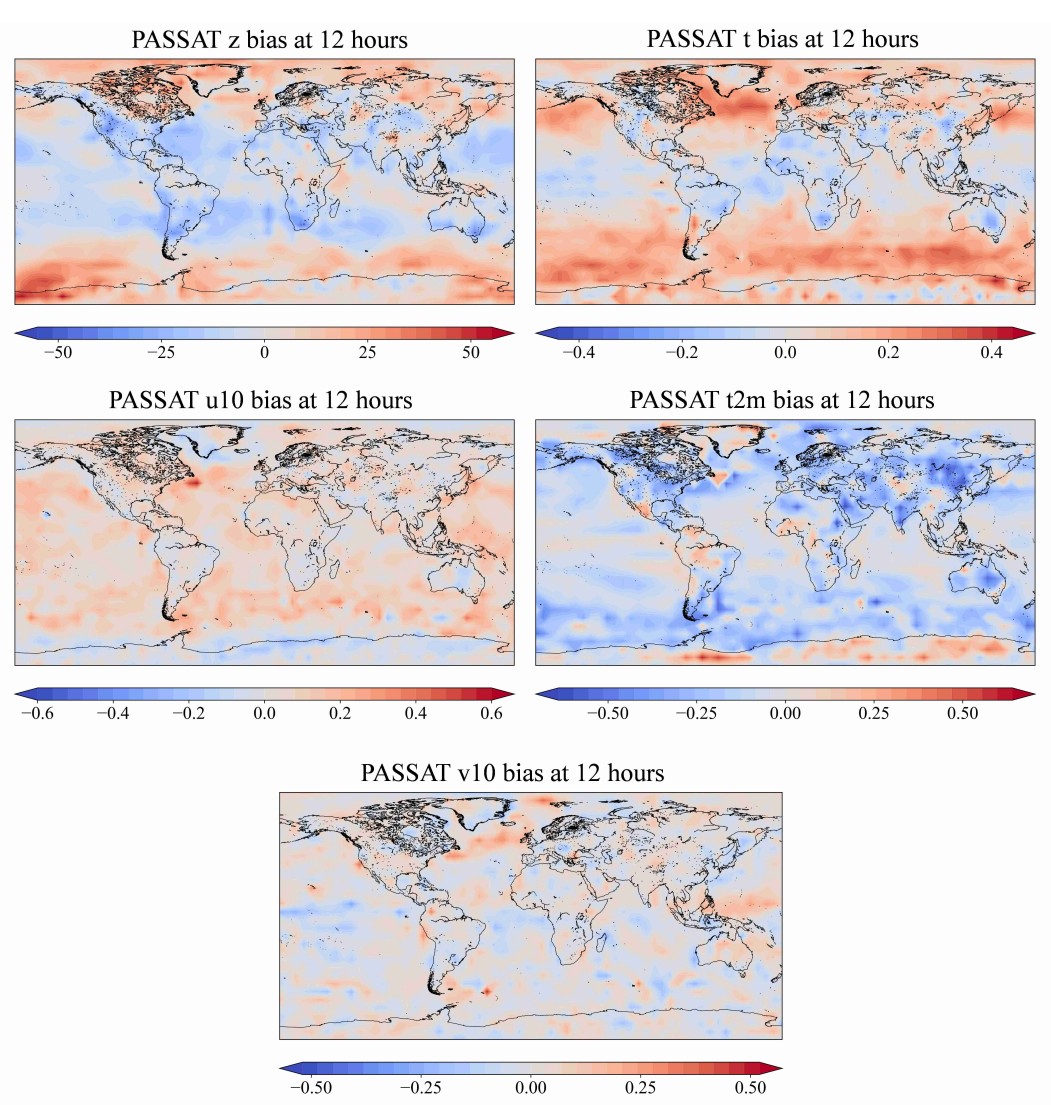

Figure 5: **MBE of PASSAT at 12 hours lead time, over the test set.**

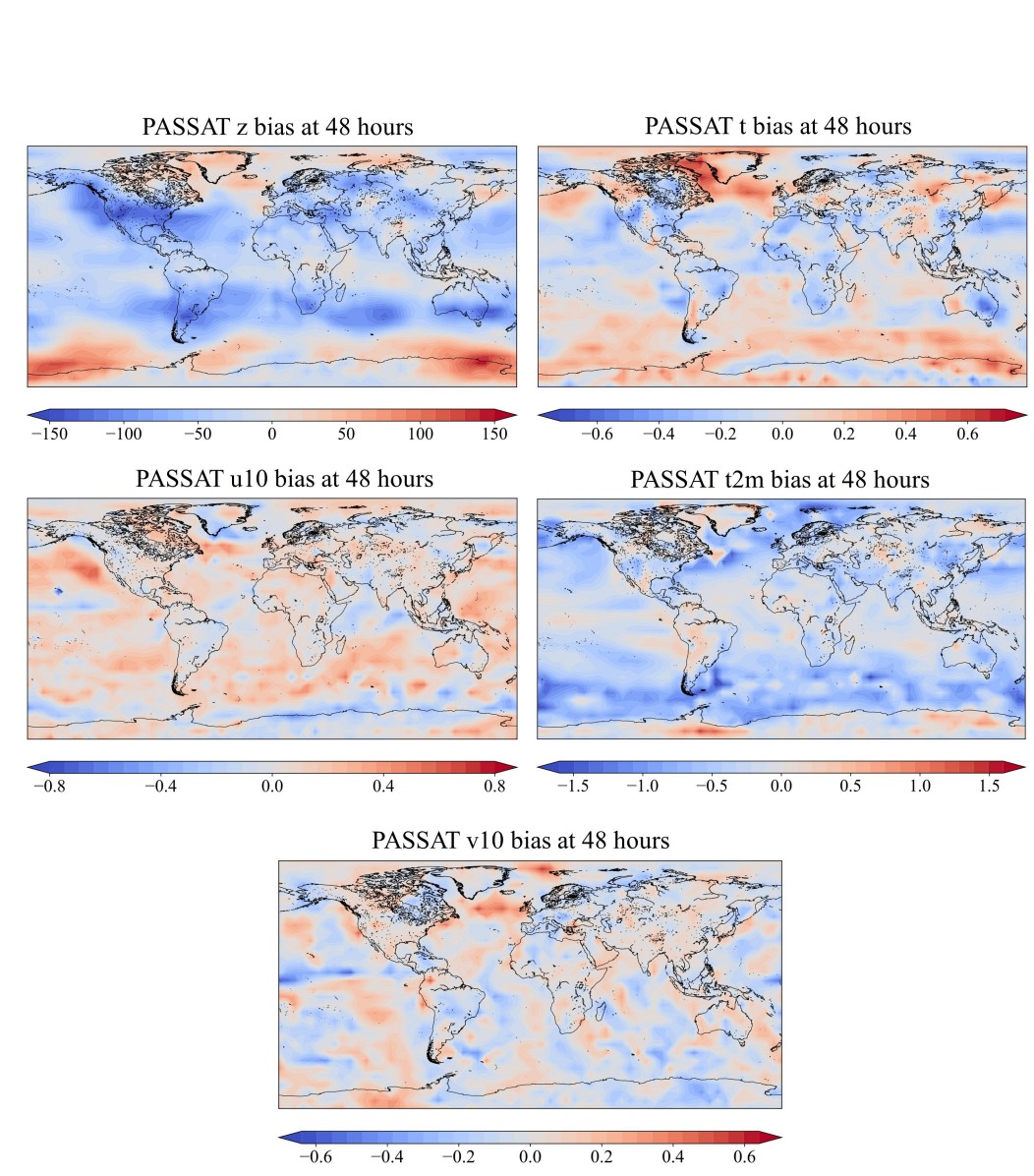

Figure 6: **MBE of PASSAT at 48 hours lead time, over the test set.**

## G.2    PREDICTIONS

We present several visualization examples of the predictions generated by PASSAT for t2m (Figure 7), t (Figure 8), z (Figure 9), u10 (Figure 10), and v10 (Figure 11).

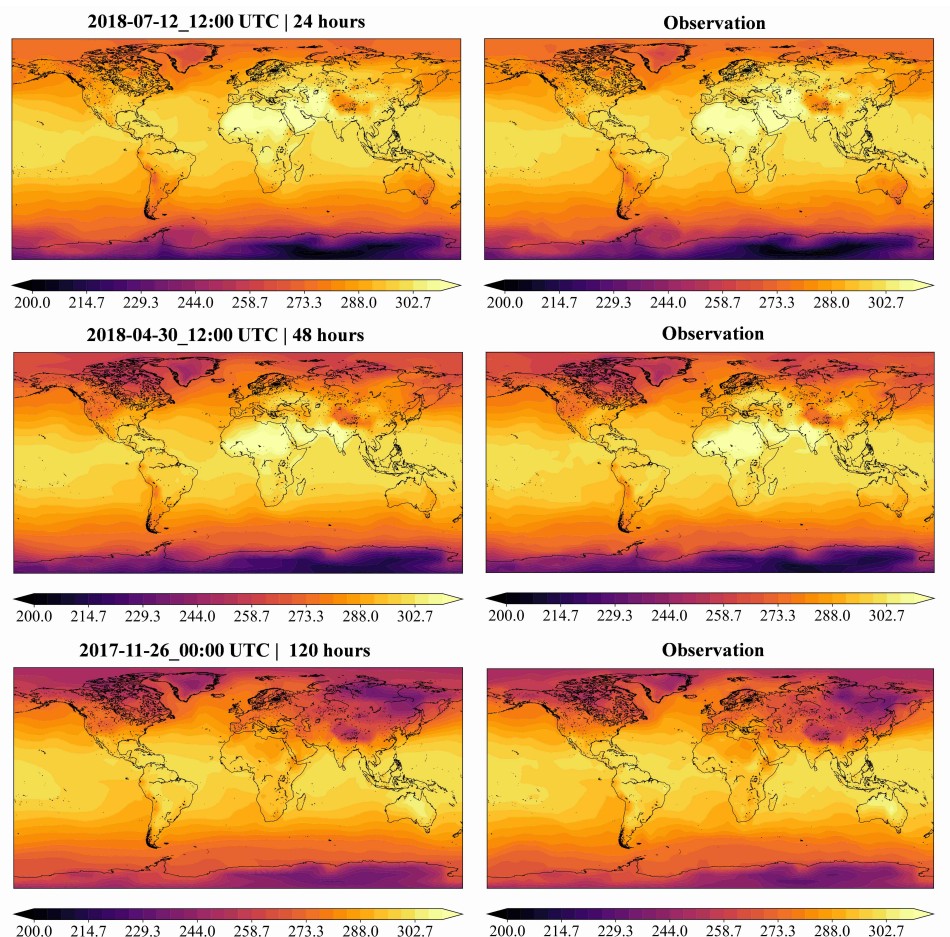

Figure 7: **Prediction visualization of t2m.**

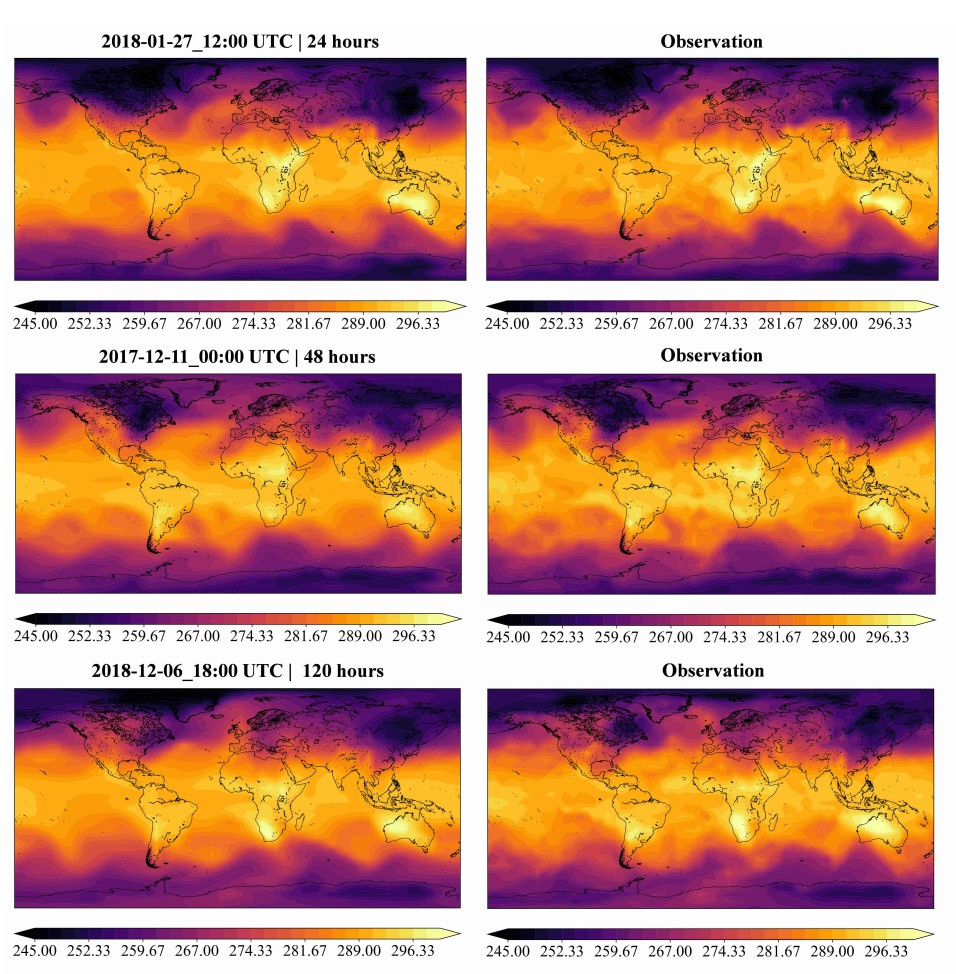

Figure 8: **Prediction visualization of t.**

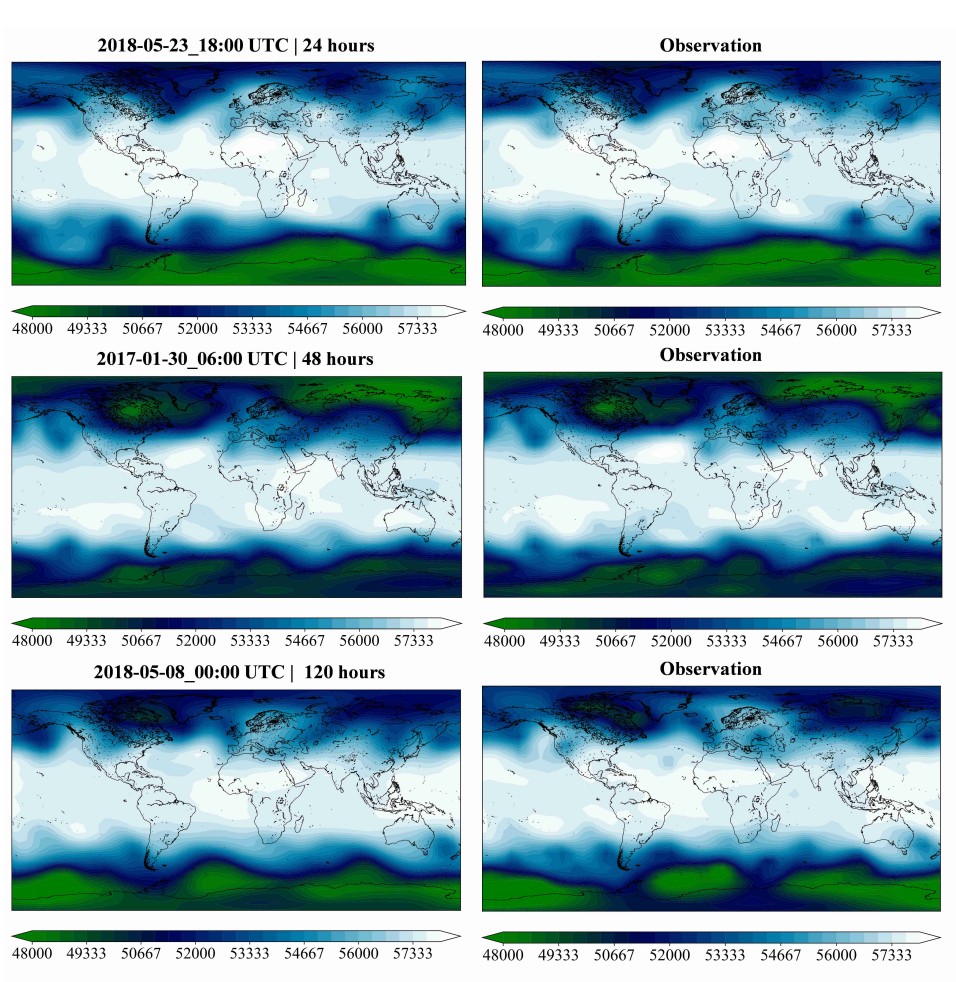

Figure 9: **Prediction visualization of z.**

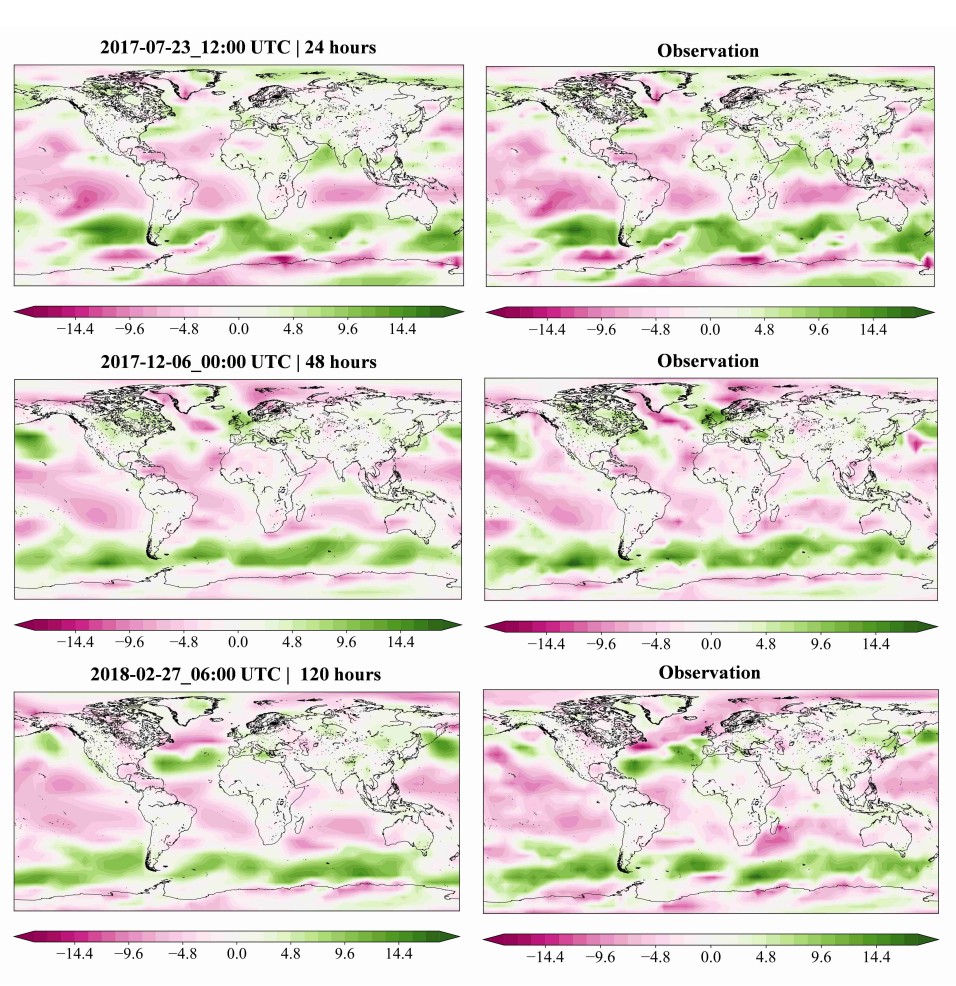

Figure 10: **Prediction visualization of u10.**

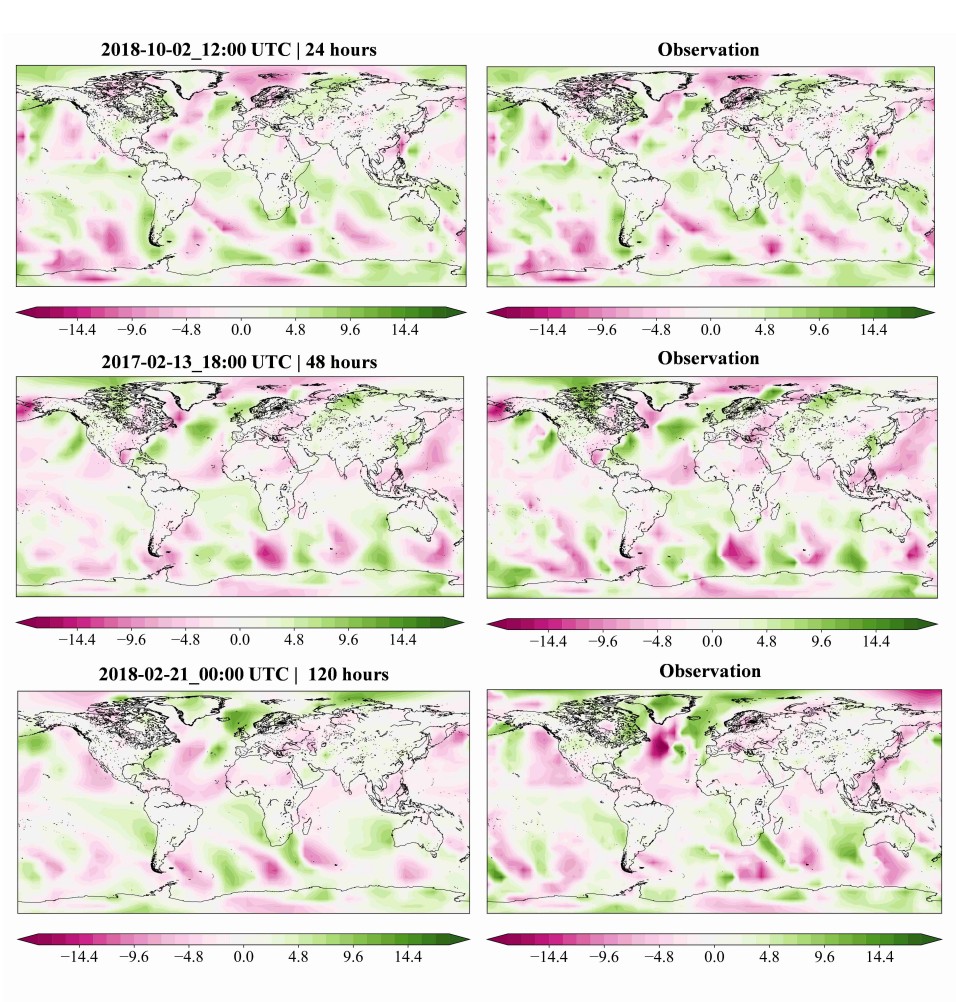

Figure 11: **Prediction visualization of v10.**

