# OpenReview forum: "Physics-Assisted and Topology-Informed Deep Learning for Weather Prediction"
_ICLR.cc/2025/Conference — Submitted to ICLR 2025_

### Official Review · Reviewer_YiMM · 2024-10-30

**Soundness:** 2
**Presentation:** 2
**Contribution:** 3
**Rating:** 3
**Confidence:** 4

**Summary:**

This paper describes a hybrid deep-learning and physics-based weather forecasting model. The authors combine the numerical solving of the advection and Navier-stokes equations with tendencies predicted by a neural network. Both components are tailored to respect the spherical geometry of earth, by adapting the physical equations accordingly and by using a spherical graph neural network for the deep learning component. Experiments are conducted on 5.625 degree ERA5 data. The proposed model shows RMSE values lower than baseline deep learning and hybrid models that do not respect the spherical geometry, and comparable values to the graph-based GraphCast* model.

**Strengths:**

1. Building hybrid deep-learning and physics-based models for the atmosphere is a highly interesting area of research, and if such models show new capabilities they can have a big impact.
2. The idea of using both the advection equation and Navier-stokes in a hybrid model like this is novel, and shows a way to take one step further in the physics direction than e.g. the ClimODE model.
3. The authors have accurately identified that many deep learning weather models do not properly handle the spherical geometry of earth. In the proposed model this is remedied in a satisfying way both in the physics equations and in the deep learning component of the model.
4. The authors experimentally compare against a relevant set of baseline models in a fair setup. The proposed PASSAT model shows strong performance over models not considering the spherical geometry.

**Weaknesses:**

**Major**:
1. My biggest doubt about this work is about the role of the physics part in the hybrid model. Looking at table 2, the version of the model w/o Navier-Stokes achieves similar or even better results. Going even further, the version of the model without the physics-component (last column, pure GNN) shows only slightly worse RMSE values than the full PASSAT model. Crucially, this pure GNN version still substantially outperforms all baselines except GraphCast*. The conclusion to draw from this is that the most important part of the model seems to be the GNN, and the authors have mainly developed a very good GNN model, rather than a model that crucially relies on the physics equations. This could of course be a contribution in itself, but is not the focus of the paper as is. There is furthermore very little description of how this GNN component of the model works, which is a shame given how important it seems to be experimentally. The framing of the paper could be changed to highlight more this contribution, but that would change the shape of the paper enough that it would warrant a resubmission.
2. The authors write in the list of contributions that "PASSAT ... remarkably improves the stability of medium-term prediction". There are no experiments to back this up. Stability is mentioned also in section 3.5, but there is no investigation of the stability of the model. The existing experiments are conducted up to 5 days, while most deep learning weather models are stable to roll out up to 10 days or beyond, so there is no support for this being an improvement in stability. Explicit experiments on how stable the different models are for long rollouts could shed some light on this.
3. I am missing important related work, that should be discussed:
  * ML Weather models handling the spherical geometry: (Bonev et al., 2023), (Esteves et al., 2023)
  * Graph-based models: (Keisler, 2022), (Lam er al., 2023), (Oskarsson et al., 2024), (Lang et al., 2024). Given that the deep learning component of the model is a GNN, I would expect some mention of existing graph-based approaches and discussion about how the model relates. Lam et al. (2023) is mentioned heavily throughout the paper, but I miss a proper discussion about how the GNN used in PASSAT relates to GraphCast.
4. The proposed model shows similar performance to the GraphCast* model. GraphCast has been reimplemented for this study, which is completely fair, but I wonder how close GraphCast* follows the original GraphCast model? In Table. 5 the number of nodes in GraphCast* is listed as 4096, which is 2 * 2048 (the number of grid points in the data). I don't see how you would arrive at this by summing the 2048 grid nodes with nodes in a multi-scale mesh graph constructed as described by Lam et al. (2023). If the graph used by GraphCast* in the paper is not such a icosahedral multi-scale graph this does not seem like a fair representation of the GraphCast model. This becomes important as GraphCast is the closest baseline competitor to the proposed method. I would appreciate if the authors could clarify this.
5. The motivation for introducing the Navier-stokes equations is unclear. This is motivated by the GNN predictions leading to error accumulation, but would not heavy error accumulation be remedied by the rollout training scheme being used?

**Minor** (not important for my final rating):
1. When discussing the importance of considering the spherical geometry, it would be a stronger argument to point to explicit problems in existing models that do not, rather than only conceptual issues. An argument can be made that flexible deep learning models can learn to correct for the problems caused by disregarding the geometry. See for example how Pangu still provides physical predictions close to the poles. An example of this not working out can be found in Bonev et al. (2023), and the discussion there is an example of a more concrete motivation for considering the geometry.
2. While I agree that existing models do not respect earth's *topology*, the fact that the earth is a sphere is even an aspect of *geometry*. I think this would be a better terminology to use.
3. The paper discusses the role of the deep learning component as modeling the earth-atmosphere interactions. I interpret earth-atmosphere interactions as interaction between the earth's surface and the atmosphere just above it. In reality, this second component of the model seems to take the role of parametrizations of all kinds of unresolved processes happening on sub-grid scales. The terminology and exact meaning of earth-atmosphere interactions should be clarified.
4. The authors write (about pure deep learning models): "Thus, their predictions are often unreliable due to the lack of the physical constraints or suffer from the distortions caused by the topological structure". I would argue that this is not an accurate statement for modern deep learning weather models, that tend to be reliably quite accurate and stable. The citation given to back this statement up is (Schultz et al., 2021), which is an earlier work than almost every single deep learning model discussed in the rest of this paper.
5. Sections 1.1 and 1.2 read as related work subsections, and would feel more at home as part of that section.
6. As opposed to what is written in the paper, GraphCast, Pangu and NeuralGCM have provided open code. Now there might be other good reasons to reimplement these for this study, but this should be clarified.


References:

* Bonev, Boris, et al. "Spherical fourier neural operators: Learning stable dynamics on the sphere." International conference on machine learning. PMLR, 2023.
* Esteves, Carlos, Jean-Jacques Slotine, and Ameesh Makadia. "Scaling Spherical CNNs." International Conference on Machine Learning. PMLR, 2023.
* Keisler, Ryan. "Forecasting global weather with graph neural networks." arXiv preprint arXiv:2202.07575 (2022).
* Lam, Remi, et al. "Learning skillful medium-range global weather forecasting." Science 382.6677 (2023).
* Oskarsson, Joel, et al. "Probabilistic Weather Forecasting with Hierarchical Graph Neural Networks." arXiv preprint arXiv:2406.04759 (2024).
* Lang, Simon, et al. "AIFS-ECMWF's data-driven forecasting system." arXiv preprint arXiv:2406.01465 (2024).

**Questions:**

1. Comparing the continuity equation used in ClimODE with the advection equation used here, the difference seems to be that you exclude the compression part. This should correspond to an assumption that the flow is incompressible. What is the motivation for this choice? Is this well-motivated for the earth's atmosphere?
2. Is my understanding correct that all equations (advection and Navier Stokes) are applied to each weather variable $u$ independently, and all interactions between them happen in the deep learning component? How can one then think of the u and v wind components? Is it reasonable to assume that these independently follow the advection equation? Not being an expert on these physical equations, my intuition would rather be that the wind components would have a close connection to the actual transport of at least the geopotential field. Could you comment on this?
3. In algorithm 1 you introduce a $\lambda$ that controls the update of the velocity field. With the value given, only $\frac{1}{60}$ weight is given to the updated value from Navier Stokes. The motivation given for this is to ensure stability. Could you expand on this choice? Is there a physical motivation for this? It seems like this could be one reason why the introduction of Navier Stokes has very little impact in the ablation study.

---

> ### Author Response · Authors · 2024-12-01
>
> Thank you for your review and suggestions. Due to time conflicts (personal reasons), we were not able to make adjustments to the manuscript in time. From your comments and those of the other reviewers, we are well aware of the shortcomings of the manuscript, and are therefore currently working on the next submission. Nevertheless, please allow me to respond to your review comments.
>
> **Major**
>
> - Major 1: Thank you very much for recognizing our work on model engineering. As you said, we hope that our contribution is to propose a new framework for combining AI and physics. However, for topological needs, we have also designed a more adapted graph neural network. We failed to properly metric the contribution of modeling vs. physics approaches. We believe that the physical framework presented in this paper is valuable, but more experiments are needed to highlight its value.
> - Major 2: Thank you for your advice. From the experimental trend, PASSAT under the NS method will be more favorable in the medium-term forecast. We will expand the forecast lead-time in the next submission.
> - Major 3: Thanks for the heads up. We will add experiments with models such as SFNO in future submissions. PASSAT differs from GraphCast in that PASSAT builds the adjacency matrix from distances between nodes in sphere and Gauss kernel, removes the multi-layer mesh structure and adds an extra graph convolution module.
> - Major 4: Thank you for your careful review. Instead of using the icosahedral and its variants to build GraphCast's graph elements, we used PASSAT's adjacency matrix A5 to build GraphCast's multi-layer mesh structure (which is probably what makes GraphCast behave similarly to PASSAT). We have recently experimented with NVIDIA's replicated GraphCast model. Surprisingly, our reproduced GraphCast* outperforms the 3-meshLevel (2690 nodes and 14392 edges) as well as the 4-meshLevel Graphcast (11782 nodes and 29732 edges).
> - Major 5: Ideally, adopting Pangu's forecasting strategy or a more aggressive forecasting strategy (one model corresponding to one forecasting step) would result in better overall performance, but with a correspondingly large training cost. Autoregressive forms of forecasting are usually the more economical choice. Compared to the model's forecast errors, we think that the errors from NS equations (resulting from modeling errors as well as numerical errors) are relatively small.
>
> **Minor**
>
> We sincerely appreciate your reminders and suggestions. Your constructive comments will help us tremendously in our next improvements.
>
> - Minor 1, 2, 3, 5:  We apologize for any distress caused to you by the terminology as well as the irregularities in the structure. Thank you for your detailed guidance.
> - Minor 4: As you said, AI Weather Forecast is accurate in the average sense and in the relevant metrics. However, they remain unreliable in the face of extreme weather conditions (e.g., the Pangu model accurately forecasts the path of typhoons, but generally underestimates their strength). We will mention related phenomena in our next contribution.
> - Minor 6: We have now completed the reproduction of Pangu as well as GraphCast through open source code, and the results are more favorable to PASSAT. Unfortunately, NeuralGCM still only provides test code, which does not allow us to train the model from scratch.

---

> > ### Author Response · Authors · 2024-12-01
> >
> > **Question**
> >
> > - Question 1: Atmospheric science usually assumes that the atmosphere is compressible. As you say, we make the assumption of incompressibility, which will facilitate the decoupling of some of the variables (e.g., atmospheric density vs. velocity). But our incompressibility assumption comes from the NS equations, not the advection equations. Rigorously, most weather variables (wind, temperature, humidity) do not contain a compression part in their dynamical kernel. The compression part has a specific meaning only when the variable is atmospheric density (continuity equation) or when it is elevation (shallow water equation). The advection part, on the other hand, is usually universal, because it describes the most basic processes that move the weather.
> > - Question 2: Thank you for your question. In our dynamic kernel, all variables are coupled to each other. Unlike ClimODE, which independently uses the continuity equations for updating and then compensates for all the errors or effects of other physical processes at once through a neural network, the dynamical system built by PASSAT takes into account the advection term as well as the sources or sinks due to other physical processes at each step of the update. The latter is obtained by inputing all variables (the state at the current moment) into the neural network. Thus, PASSAT does not update each variable independently. We need to clarify that we have not assumed that the weather variables satisfy the advection equation, which is clearly wrong. We mention the advection equation in order to obtain the advection term in the update equation, which only forms part of the trend of the weather variables.
> >
> >     The u-winds and v-winds are good estimates of velocity, but we believe that the velocity of different weather variables may still differ. This is what drives our modeling of the velocity field. The influence of geopoential to winds is given by neural network.
> >
> > - Question 3: Numerical weather forecasting uses a number of filters to maintain the stability of the numerical results. Relatively, we would like to realize this thing by convex combination. However, as you mentioned, too small convex combination coefficients make the NS equations produce a smaller positive impact. We will explore this parameter setting more carefully in subsequent experiments.

---

### Official Review · Reviewer_mMof · 2024-11-02

**Soundness:** 3
**Presentation:** 2
**Contribution:** 2
**Rating:** 5
**Confidence:** 3

**Summary:**

This paper introduces a new deep learning model PASSAT (Physics-ASSisted And Topology-informed) for weather prediction tasks. It first attributes the weather evolution to two major factors which are the advection process (characterized by the advection equation and the Navier-Stokes equation) and the Earth-atmosphere interaction. PASSAT works by training a spherical graph neural network to estimate interactions between the Earth and atmosphere and use it to create initial velocity fields. It also solves the advection equation on the sphere and update velocity fields by solving the Navier-Stokes equation on the sphere. With this certain methods, PASSAT reduces the uncertainty brought by cumulative errors to mid-term forecasts and increase forecast accuracy. Experiments show that PASSAT outperforms existing deep learning and traditional numerical weather prediction models on the ERA5 dataset, especially in terms of stability and accuracy of medium-term forecasts.

**Strengths:**

1. Combining physical and data-driven methods at the same time: PASSAT model uses both physical equations (convection equations and Navier-Stokes equations) and deep learning method (spherical graph neural network) to make the prediction results more accurate, robust and more interpretable, and reduces the cost as well as the model uncertainty compared to purely data-driven methods.

2. Considering the spherical topology: The model innovatively designs a spherical graph neural network to perform calculations on the sphere, avoiding the errors caused by using plane projection, which improves the accuracy of global weather forecasts.

3.The experimental verification effect is significant: Experimental results on the ERA5 data set show that the PASSAT model surpasses many current advanced models in medium-term weather prediction under similar parameter quantities, demonstrating its potential in practical applications.

**Weaknesses:**

1. The author's choice of the number of variables is not good enough. Although the author mentioned this shortcoming, it cannot be ignored. For example, FourCastNet predicts 20 climate variables at the same time. Predicting multiple variables at the same time will inevitably have a certain impact on the performance of the model. How does the author view this impact?

2. Due to the timeliness of climate prediction, the training time and inference time of the model are also very important evaluation indicators. The author mentioned that PASSAT has the characteristic of less computational complexity. Can you give a specific comparison of training time and inference time?

3. The author mentioned "To be specific, we no longer trust the velocity field v estimated by the velocity branch of the spherical graph neural network, except for the initial time t. Instead, we solve the Navier-Stokes equation that governs the evolution of the velocity field, to calculate v." So how exactly does the author "no longer trust the velocity field v"?

4. An extra ablation experiment can be designed to further demonstrate the difference between using a spherical graph neural network and a normal planar graph neural network.

5. Appendix E mentions how the authors compressed the baseline model, but the authors need to provide more detailed evidence to prove that such compression is reasonable and fair, especially for baselines such as GraphCast and Pangu whose original model parameters are relatively large.

**Questions:**

For the questions, please see the weakness section. If you can answer my question, I will improve my rating.

---

> ### Author Response · Authors · 2024-12-01
>
> Thank you for your review and suggestions. Due to time conflicts (personal reasons), we were not able to make adjustments to the manuscript in time. From your comments and those of the other reviewers, we are well aware of the shortcomings of the manuscript, and are therefore currently working on the next submission. Nevertheless, please allow me to respond to your review comments.
>
> - W1: Thank you for your question. In our experimental setup, all models predict 5 variables simultaneously. Comparing with FourCastNet's experiments, the setup in this paper is indeed too simple. However, from the point of view of fairly comparing the performance of different models, we think this is sufficient to illustrate the superiority of PASSAT.
> - W2: We apologize for any distress caused by what we said. As we mention in Section 1.1,  physics-assisted deep learning models are harder to train and slower in inference compared to the end-to-end deep learning methods. Our statement “PASSAT has the characteristic of less computational complexity” is refer to comparison between PASSAT and numerical weather prediction models.
> - W3: Thank you for your rigorous and careful review. We got this conclusion from the experimental results (for 120-hour forecasts, the results using the NS equation would be relatively better).
> - W4: Thank you very much for your advice. We will improve it for the next submission.
> - W5: Your question is to the point. The experimental setup for this manuscript is relatively simple. GraphCast, Pangu, and FourCastNet at the original parameters can overfit quickly, resulting in their performance far worse than a model with 1M parameter size.

---

### Official Review · Reviewer_eZ3m · 2024-11-03

**Soundness:** 2
**Presentation:** 2
**Contribution:** 1
**Rating:** 3
**Confidence:** 4

**Summary:**

This paper introduces PASSAT, a physics-assisted and topology-informed deep learning model designed for weather forecasting. PASSAT employs a graph neural network to anticipate the velocity field associated with interaction and advection processes, followed by the application of numerical methods forecast the velocity field and other variables using the advection equation and the Navier-Stokes equation, respectively.

**Strengths:**

PASSAT incorporates the physical processes inherent in weather forecasting. It also addresses the discrepancies between the differences between equiangular observations from ERA5 data and planer expansion typically assumed in the existing models, offering a more accurate representation for the real world.

**Weaknesses:**

1. The experiment only utilized T2m, 10u, 10v, Z500, T850 from ERA5 as input and target variables, leading to two major problems:
   - These variables are not at the same altitude, which impedes the accurate application of the Navier-Stokes equation to velocity and temperature observations at varying heights. The ablation experiment further indicated that introducing the Navier-Stokes equation did not significantly improve the prediction.
   - Regarding the incorporation of temperature, existing methods of embedding physical equations such as NeuralGCM[1] consider the thermal dynamics introduced by temperature variations and employ the more commonly accepted primitive equations for physical modeling.
2. The author only used 13 years of ERA5 data with a spatial resolution of 5.625° for their experiment, which is a relatively simple prediction task. It is recommended that the authors use the highest resolution data of 1.40625° available in WeatherBench to enrich their dataset.
3. When compared to GraphCast, the proposed method hdemonstrated a marginal improvement of approximately 1%. In the ablation experiment, the inclusion of Navier-Stokes equations and advection equations resulted in only a modest  performance promotion, which does not validate the effectiveness of the author's design.

[1] Neural General Circulation Models for Weather and Climate. https://arxiv.org/abs/2311.07222

**Questions:**

See weaknesses above.

---

> ### Author Response · Authors · 2024-12-01
>
> Thank you for your review and suggestions. Due to time conflicts (personal reasons), we were not able to make adjustments to the manuscript in time. From your comments and those of the other reviewers, we are well aware of the shortcomings of the manuscript, and are therefore currently working on the next submission. Nevertheless, please allow me to respond to your review comments.
>
> - W1: On the one hand, we do not use the NS equations to update the temperature, but rather the velocity field; on the other hand, since the velocity field is an output of the deep learning model, even if the NS equations are not completely accurate, the model can adjust the initial velocity field based on the data as well as the loss function;
> Our manuscript has references to NeuralGCM. Considering the limitation of computational resources, we are not able to consider fully primitive equations. Therefore, we only numerically compute the advection term and use the spherical GNN to learn the other resolved or unresolved physical processes.
> - W2: Thank you for your advice. We currently recognize this as a key issue as well. We will rework the dataset in subsequent submissions.
> - W3: We think it makes sense that with **fair parameters and settings**, PASSAT can perform better than GraphCast for most variables and most forecast durations.

---

### Official Review · Reviewer_MqVr · 2024-11-04

**Soundness:** 3
**Presentation:** 2
**Contribution:** 3
**Rating:** 3
**Confidence:** 3

**Summary:**

This paper introduces PASSAT (Physics-ASSisted And Topology-informed), a deep learning model designed to improve weather prediction by integrating physical and topological information. PASSAT combines physics-driven PDEs (such as the advection and Navier-Stokes equations) with a spherical graph neural network (Spherical GNN) that leverages Earth’s topology. Experiments on the ERA5 dataset demonstrate that PASSAT outperforms several state-of-the-art weather forecasting models, including FourCastNet, Pangu, and GraphCast.

**Strengths:**

The model demonstrates superior performance on the ERA5 dataset compared to leading weather forecasting models.

**Weaknesses:**

1. The architecture of PASSAT, particularly the internal workings of the spherical GNN and the interaction between the “interaction branch” and “velocity branch,” is not sufficiently detailed. A more detailed architectural diagram and step-by-step explanation would make the model’s structure and operation easier to understand and potentially reproduce.

2. The combination of a spherical GNN and Navier-Stokes equations implies high computational demands. The paper does not provide benchmarks for computational efficiency, which is crucial for real-world applications.

3. The benchmarks should include the SOTA Numerical Weather Prediction (NWP) models, which are the industry standard for accuracy and robustness in weather forecasting.

4. While Navier-Stokes equations are employed to capture fluid dynamics on a global scale, the work does not clarify how it manages varying boundary conditions across different geographical or atmospheric regions.

5. **(Major concern)** Although the paper employs the Navier-Stokes (NS) equations for physical realism, it does not provide a theoretical justification for their selection over other possible fluid dynamics models. Moreover, it is unclear how PASSAT addresses the variability and abrupt changes often observed in real-world data, which may not always align with the idealized assumptions of the NS framework. For instance, sudden atmospheric shifts and noise present in observational data could challenge the applicability and robustness of the NS-based approach in capturing highly dynamic or turbulent weather events.

**Questions:**

See weakness.

---

> ### Author Response · Authors · 2024-12-01
>
> Thank you for your review and suggestions. Due to time conflicts (personal reasons), we were not able to make adjustments to the manuscript in time. From your comments and those of the other reviewers, we are well aware of the shortcomings of the manuscript, and are therefore currently working on the next submission. Nevertheless, please allow me to respond to your review comments.
>
> - W1: Thanks for the heads up. We have a complete description of the Basic Block (including three operations on the spheircal surface) of the spherical GNN in Appendix B.2, but may not have described its relationship with the Backbone model and the Branch model in detail. The Backbone model and the two Branch models consist of multiple Basic Blocks of the spherical GNN. The input data will pass through the Backbone model first, and then the output of the Backbone model will go to the two Branch models at the same time to get the final result.
> - W2: As we mention in Section 1.1,  physics-assisted deep learning models are harder to train and slower in inference compared to the end-to-end deep learning methods. Therefore, your concern is correct. But most of physics-assisted deep learning models are still greatly computational efficient compared with numerical weather prediction methods, since they (including PASSAT) only include part of the primitive equations, or their linear approximation.
> - W3: Thank you for your advice. Since all of models in our experiment are currently unable to outperform the IFS T63 at 5.625° spatial resolution, we have neglected to include the operational IFS as a comparison. We will re-add it in a future version.
> - W4: In our opinion, there are no conditions at the boundary of the latitude-longitude domain except the obvious conditions resulting from periodicity. It only remains to specify the conditions at the boundary of the altitude or pressure levels. In short, of the variables we consider, only t2m may be affected by the bound. However, for the convenience of algorithm parallelism, we still treat them as variables within the open kernel of the solution domain.
> - W5: Thank you for your comments. As you mentioned in Weakness 2, we had to consider simplified NS equations (e.g., shallow water equations) in order to balance computational efficiency. Since we found in most of the research related to primitive equations that people tend to use incompressible NS equations for modeling as well as analysis, we have chosen the incompressible NS equations directly.
> As for your second question, we believe that it is not possible to have a completely realistic model. Even the most advanced numerical weather prediction models have subgrid processes and microphysical processes that cannot be reasonably modeled. At least in our experiments, good accuracy is still achieved using the NS equations.

---

### Meta-Review · Area_Chair_YB2g · 2024-12-19

**Metareview:**

The paper introduces a novel hybrid DL model for weather prediction that takes into account the underlying physics (characterized by the NS equation) as well as the geometry of the earth (through the spherical GNN). The paper has experiments on ERA5 that shows the new model outperforms other SOTA DL models.

Strengths: novel idea in combining physics through the NS PDEs into solving a very important problem
Weaknesses: Limited evaluation (results are on coarse ERA5 and hence not very reliable), insufficient evidence that show the value of adding the physics itself, insufficient variables chosen to model atmospheric physics

Adding more experimental evidence for the value of the physics, making fairer comparisons on the high-resolution dataset (all SOTA weather models today are trained on 0.25 degree ERA5 data) will make the paper stronger.

**Additional Comments On Reviewer Discussion:**

The reviewers mainly raised concerns detailed in the weaknesses above. While the idea is novel and very interesting, there is insufficient experimental results to validate the authors's hypothesis.
Other technical concerns were raised. The authors were unable to make necessary changes to strengthen their paper in the rebuttal due to personal reasons.

---

### Decision · Program_Chairs · 2025-01-22

Reject